



# Advanced method for estimating the number concentration of cloud water and liquid water content observed by cloud particle sensor sondes

Jun Inoue[1,2], Kazutoshi Sato[3], Yutaka Tobo[1,2], Fumikazu Taketani[4], and Marion Maturilli[5]

[1]National Institute of Polar Research, Tachikawa, 190-8518, Tokyo, Japan
[2]The Graduate University for Advanced Studies, SOKENDAI, Tachikawa, 190-8518, Tokyo, Japan
[3]Kitami Institute of Technology, Kitami, 090-8507, Hokkaido, Japan
[4]Japan Agency of Marine-Earth Science and Technology, Yokohama, 236-0001, Kanagawa, Japan
[5]Alfred Wegener Institute, Helmholtz Centre for Polar and Marine Research, Potsdam, 14473, Germany

**Correspondence:** Jun Inoue (inoue.jun@nipr.ac.jp)

**Abstract.** A cloud particle sensor (CPS) sonde is an observing system attached with a radiosonde sensor to observe the vertical structure of cloud properties. The signals obtained from CPS sondes are related to the phase, size, and number of cloud particles. The system offers economic advantages including human resource and simple operation costs compared with aircraft measurements and land-/satellite-based remote sensing. However, because CPS systems are limited for data downlink to land stations, the observed information should be appropriately corrected. We launched approximately 40 CPS sondes in the Arctic region between 2018 and 2020 and use these data sets to develop correction methods that exclude unreliable data, estimate the effective cloud water droplet radius, and determine a correction factor for the total cloud particle count. We apply this method to data obtained in October 2019 over the Arctic Ocean and March 2020 at Ny-Ålesund, Svalbard, Norway to compare with a particle counter onboard a tethered balloon and liquid water content retrieved by a microwave radiometer. The estimated total particle count and liquid water content from the CPS sondes generally agree with those data, which exemplifies the promising advantages of this approach to retrieve quantitative and meaningful information on the vertical distribution of cloud microphysics.

## 1 Introduction

Clouds regulate weather and climate systems by radiation, precipitation, and the transfer of heat and moisture (Boucher et al., 2013). They cover a large part of the earth and range in scale from tens to thousands of kilometers (e.g., cyclones or frontal systems). However, the formation process of cloud droplets occurs on the micro-scale through complicated cloud microphysical processes. The simulation of clouds in general circulation models (GCMs) therefore depends on numerous parameterizations.



The composition of clouds involves a liquid phase (including supercooled), solid-phase, or mixed phase. There are numer-
ous microphysical processes related to the formation of cloud water/ice and falling hydrometeors. In the temperature range
between 0°C and approximately −40°C, cloud particles may exist both as liquid and ice (Korolev et al., 2017). Although cloud
phases and their vertical and horizontal distribution are critical to calculating downward shortwave and longwave radiation, the
representation of clouds in climate models, including the partitioning of liquid and ice in clouds, remains a challenging issue
because of the poor understanding of cloud microphysical processes.

The partitioning of ice and liquid in mixed-phase clouds controls the radiation budget at the Earth's surface, with particular
implications to the vulnerable ice-ocean system of the high latitudes, where the radiative interactions between the microphysical
and macrophysical properties of clouds and the surface modify the warming or cooling effect of clouds (Stapf et al., 2020). An
overestimation of cloud ice, which has less shortwave radiation reflection, tends to generate a positive sea-surface temperature
(SST) bias in the Southern Ocean (Flato et al., 2013). The representation of long-lasting clouds (i.e., cloud water) instead of
cloud ice is critical to reducing such SST bias in the Southern Ocean (Varma et al., 2020). Moreover, cloud ice/water fractions
in the models represent the air-sea coupled system, which includes ocean circulation and changing winds induced by corrected
temperature gradients (Kay et al., 2016).

In the Arctic region, the surface energy budget, particularly over sea-ice, is constrained by shortwave radiation during
summer and longwave radiation during winter (Intrieri et al., 2002; Persson et al., 2002). The representation of clouds and their
impact on radiation fields are therefore vital to simulate future Arctic climate systems. A comparison of regional climate models
shows that cloud water is poorly represented in these models (Sedlar et al., 2020). Based on ice-free Arctic Ocean results,
Inoue et al. (2020) found that the double-moment cloud microphysics scheme, which solves the mixing ratio and number
concentration for each hydrometeor, is superior to the single-moment cloud microphysics scheme. One of the remaining issues
is that ice-nucleating particles should be carefully tuned for target seasons/locations when the double-moment cloud physics
scheme is applied (Inoue et al., 2020).

Satellites have provided a global perspective of clouds and radiation (Stubenrauch et al., 2013) and satellite data have
been intensively used for atmospheric reanalysis (Hersbach et al., 2020). With the advancement of satellite data in recent
years as well as computational resources, the presence of global cloud-resolving models (GCRMs), which resolve both large-
scale dynamics and small-scale convection, has increased in global weather and climate simulations (Satoh et al., 2019) and
global precipitation forecasting with the aid of data assimilation (Kotsuki et al., 2019). Although a general agreement of
modeled clouds using satellites in some deep cloud development processes has been reported, GCRMs still depend on cloud
microphysical parameterizations, such as high thin cirrus parameters (Kodama et al., 2012).

Despite these advances, the size distribution of cloud particles and vertical distribution of cloud mixing ratios remain poorly
validated using observations for boundary layer clouds. Cloud phases can be confirmed by land-/ship-based remote sensing
such as long-term monitoring using cloud radars and microwave radiometers (Illingworth et al., 2007; Nomokonova et al.,
2019), by a cloud particle imager onboard a tethered balloon system (Lawson et al., 2011), and by a fog monitor at the top of
mountain (Koike et al., 2019); however, the number concentration of clouds is more difficult owing to observation logistics.
For example, aircraft observations are costly, which limits the number of feasible flights, and tethered balloons are weather-





dependent (i.e., wind speed) and thus limited to a top height of approximately 1000 m. Unknown particle collection efficiency
is also a problem for quantitative understanding. The cost and mobility of observation data are important aspects that should
complement existing observation systems, including satellites.

A cloud particle sensor sonde (Meisei Electric, Co., Ltd.; hereafter, CPS sonde) is an observation system used to obtain the
vertical profile of cloud information (e.g., total particle count, particle phases, and particle size) (Figs. 1, 2). A CPS connected to
a normal radiosonde can obtain cloud parameters and basic meteorological profiles. The observation cost consists of the regular
launch of a radiosonde and an additional $ 1200 for the CPS. Although theoretical configurations and laboratory experiments
have been intensively reported (Fujiwara et al., 2016), the data require adequate corrections adapted to the individual flight
dynamics. The remaining issues are: (1) the relationship between flow speed in the CPS inlet and CPS signal; (2) a theoretical
understanding of the time interval of each particle signal; (3) characteristics of the aerodynamic flow pattern around the CPS
housing, which determines the sampling volume; and (4) validation of the CPS sonde with other observation systems. In this
study, we propose a CPS data correction method using the observation data obtained during three Arctic field campaigns (Fig.
1, Table 1) and idealized simulations. The corrected cloud parameters are validated by other observation data sets.

## 2   Overview of a CPS sonde system and its remaining issues

### 2.1   Particle detection

The technical details of a CPS are described in Fujiwara et al. (2016) from which the essential features are introduced here. A
CPS uses a near-infrared laser with a typical 790-nm wavelength as a linearly polarized light source. Two silicon photodiodes
are placed at angles of $55°$ (detector no. 1) and $125°$ (detector no. 2) to the source light direction (Fig. 2). A polarization plate
is placed in front of detector no. 2 so that it only receives light polarized perpendicular to the light source. The two detectors
receive scattered light through the slits (0.50 cm × 1.0 cm) in front of them. Although the detection domain is estimated as $\sim$
1 cm × 1 cm × 0.5 cm, because the two detectors collect light scattered at $55°\pm10°$ and $125°\pm10°$ (Fujiwara et al., 2016), we
75  speculate the critical detectable domain for each detector should be thinner than previously proposed (i.e., <0.5 cm). Based
on each detector's angle and the light source thickness through a slit (0.5 cm), the domain shape for each detector should be a
parallelogram, as shown in Fig. 3a. This non-uniform light source thickness for each detector allows the cloud particles to pass
through the zone over different detection times, defined as the particle signal width (PSW: unit in ms), even if the flow speed
and particle size are the same.

80  The particle signal voltage from the two detectors ($I_{55}$ and $I_{125p}$) range from 0 to 7.5 V with a resolution of 0.03 V. Based on
laboratory experiments performed to determine the lower particle size detection limit and relationship between $I_{55}$ and water
droplets, the CPS cannot detect 1-$\mu$m diameter polystyrene particles but can detect 2-$\mu$m diameter borosilicate glass particles
(Table 2) (Fujiwara et al., 2016). The CPS often gives saturated outputs ($\sim$7.5 V) for 60- and 100-$\mu$m diameter soda-lime glass
particles. In this study, we construct an experimental equation to estimate the liquid cloud effective radius from the measured
85  voltages, as described in section 4.4.



## 2.2 Limitation of data down link

Owing to the downlink capability of the Meisei radiosondes, only 25 byte s$^{-1}$ can be transferred to the ground-based receiver. The current CPS provides the following information each second: (i) number of particles (particles s$^{-1}$); (ii) CPS signal voltage for $I_{55}$ and $I_{125p}$ (V) and PSW (ms) for the first six particles entering the instrument each second; and (iii) DC component for the detector no. 1 output. It should therefore be noted that it is impossible to obtain the particle size distribution every second. A statistical approach is necessary to estimate the liquid water content (LWC) and liquid water path (LWP).

## 2.3 Detection time for each particle

PSW is considered to be an indicator of particle transit time when $I_{55}$ first exceeds 0.3 V and the time when $I_{55}$ falls below 0.3 V. According to Fujiwara et al. (2016), a 5 m s$^{-1}$ flow speed corresponds to a PSW of ~1 ms for a single particle. They also reported that PWS data can be used to monitor potential particle overlap in dense cloud layers. An excessively long signal width may indicate the overlapping of too many particles in the detection area and thus a substantial loss of particle counts. In such cases, multiple light scattering can also occur and complicate the particle measurements. As described in section 2.1, however, the detection cross-section is not uniform but rather a parallelogram for each detector with a maximum thickness of 0.5 cm, which allows a wide range of PSWs even under a constant flow speed in the CPS inlet. Fig. 3b reveals the idealized PSW distribution under a flow speed of 5 m s$^{-1}$ assuming that the shape of the detectable cross-section of detector $I_{55}$ is a parallelogram. The PSW value can vary from 0 to 1 ms. Furthermore, if the flow speed slows near the inlet wall owing to frictional forces, the PSW might be large because of the time required to pass the detection area (i.e., slower flow speeds require longer times).

## 2.4 Total particle count

Fujiwara et al. (2016) roughly estimated that the CPS can correctly measure number concentrations up to ~1000 particles s$^{-1}$ under a flow speed ($v$) of 5 m s$^{-1}$. In the case of dense clouds, the PSW might be larger than 1 ms owing to signal overlap and thus lose particle counts. Fujiwara et al. (2016) therefore proposed a correction factor ($f$) for the total count of particles per second as $f = 4 \times (PSW/(5/v))^3$ if the PSW, which is the maximum among up to six values per second, is greater than $5/v$; if the PSW is smaller than $5/v$, $f = 1$. The corrected count $N_{cor}$ (s$^{-1}$) can thus be estimated using $f$ and the original count $N_{org}$ (s$^{-1}$) as follows.

$$N_{cor} = f \times N_{org} \tag{1}$$

However, this assumption is only be applicable when the detection area depth is assumed to be uniform in the CPS inlet; however, as described in section 2.3, the PSW will vary widely owing to the inhomogeneous detection cross-section domain (Fig. 3). This correction method should therefore be validated by other observed products.



## 2.5 Detection of particle phases

To distinguish between cloud ice and cloud water, the degree of polarization (DOP) is defined by Fujiwara et al. (2016) as follows.

$$DOP = \frac{I_{55} - I_{125p}}{I_{55} + I_{125p}} \tag{2}$$

When the DOP value is negative, the particle is ice. When the DOP value is positive but less than ~0.3, the particle is most likely ice. When the DOP value is more than ~0.3, the particle is water in many cases, but there is a chance that it may be ice because the DOP can take values between −1 and +1 for ice particles. In this study, a stricter DOP threshold of 0.5 is used to focus on the cloud water droplet variables.

## 3 Experimental designs

### 3.1 Field experiments in the Arctic regions

The Arctic research cruise was undertaken by the Japanese ice-strengthen Research Vessel (RV) Mirai in the Chukchi Sea in November 2018 (Inoue, 2018) (Fig. 4a). This polar night cruise provided favorable conditions for CPS sondes because strong sunlight affects the measurements of scattered light (Fujiwara et al., 2016). The observed area is the marginal ice zone. The total number of observations was 12, of which 6 are used in this study (liquid cloud case). A similar cruise was made in October 2019 (Sato, 2019) (Fig. 4b) in which the observations were made mainly at night. The total number of observations was 12, of which 11 flights are used in this study (Fig. 1a). In addition to the normal CPS sonde observations, the CPS sonde was also applied to onboard tethered balloon observations using an airship-shaped balloon (15 m$^{-3}$, The Weather Balloon MFG, Co., Ltd.). An instrument bag and CPS sonde were respectively connected 5 and 10 m below the balloon (Fig. 5). One of the instruments in the bag used in this study was an optical particle counter (OPC; HHPC6+, Beckman coulter), which has six channels for particle size ranges of 0.3–0.5, 0.5–1, 1–2, 2–5, 5–10, and >10 $\mu$m. Three channels for sizes >2 $\mu$m were used to validate the total particle counts by the CPS sondes. The ascending speed was typically 0.5–1 m s$^{-1}$, which strongly differs from the normal CPS sonde observations; however, the impact of ascending speed on particle counting is confirmed by numerical simulation, as discussed later. The maximum height of each flight was lower than 1000 m. Three cloudy cases are investigated in this study.

The field campaign based in Ny-Ålesund, Svalbard, Norway, was made in March 2020 (Figs. 1b, 4c). The surface air temperature was approximately −20 °C, which is ~10 °C lower than the previous two cruises. We selected 5 cases out of a total of 14 flights. The LWP was monitored at the French–German Arctic research base AWIPEV at Ny-Ålesund, Svalbard, which includes the Alfred Wegener Institute Helmholtz Centre for Polar and Marine Research (AWI) and French Polar Institute Paul Emile Victor (PEV). At the AWIPEV station, a humidity and temperature profiler (HATPRO), a passive microwave radiometer, has been in operation since 2011 (Rose et al., 2005). Nomokonova et al. (2019) reported that more than 90% of single-layer liquid clouds have LWP values lower than 100 g m$^{-2}$. The Cloudnet product (archived at http://devcloudnet.fmi.fi/) contains an





adiabatic retrieval of LWC when both the cloud radar and lidar detect a liquid layer, and microwave radiometer data are present (most reliable classification). In this study, the LWP data and LWC retrieved by the Cloudnet product are used for comparison with the CPS sonde results.

All soundings consisted of a CPS with Meisei radiosondes (RS-11G). The Vaisala radiosonde (RS41-SGP) was also simul-
taneously launched by by hanging from the opposite side of a 1-m-long rod in the 2019 and 2020 campaigns (the other side was used for the CPS sonde) (Fig. 1). The balloon-type was a 350-g balloon (TOTEX TA350) in the 2018 and 2019 campaigns and a 600-g balloon (TOTEX TA600) in the 2020 campaign. The detailed data list is shown in Table 1.

## 3.2    Numerical experiments

The simplified three-dimensional computational fluid dynamics (CFD) was simulated using Flowsquare$^+$ software (Nora Sci-
entific, https://fsp.norasci.com/en/) to better understand the flow pattern around the CPS housing. Flowsquare$^+$ solves for density, velocity, temperature, mass fraction of a substance in the fluid, and pressure. The model domain was set to 201 (x) $\times$ 201 (y) $\times$ 601 (z) grids with 1.5 mm in horizontal resolution ($\Delta x = \Delta y$) and 1.0 mm in vertical resolution ($\Delta z$). An inflow boundary was specified at the upstream side of the z-direction. The remaining boundaries were treated as outflow boundaries under open boundary conditions. The simple 3D object assuming the CPS housing (100 mm $\times$ 100 mm $\times$ 120 mm) with the
Meisei RS-11G radiosonde was used for the inner model boundary condition. The CPS inlet was set to the CPS housing center with 10 mm $\times$ 10 mm. Flowsquare$^+$ monitors the maximum Courant-Friedrichs-Lewy number and dynamically adjusts the time step size. The air temperature and pressure were set to $-20.0\,^{\circ}$C, and 850 hPa, respectively, assuming the field campaign in March 2020 at Ny-Ålesund. The dynamic viscosity $\mu$ was fixed at $1.6 \times 10^{-5}$ kg m$^{-1}$ s$^{-1}$. The inflow speed was set to 5 m s$^{-1}$ assuming an ascending speed (exp-5m), whereas the horizontal wind speed was fixed to 0 m s$^{-1}$.

Three types of sensitivity experiments were made: (1) ascending speeds of 4 and 6 m s$^{-1}$ for typical soundings (exp-4m/6m) and 1 m s$^{-1}$ for the tethered balloon measurements (exp-1m); (2) ambient air pressure of 50, 100, 300, 500, 700, and 925 hPa (exp-50hPa/100hPa/300hPa/500hPa/700hPa/925hPa), and (3) horizontal wind speeds with 2.5 and 5 m s$^{-1}$ (exp-h2.5m/h5m). In each experiment, a time integration with 4000 steps was made, corresponding to the typical physical time scale of approximately 0.1 s, which is sufficient when the air mass at the initial state passes across the model domain (i.e.,
quasi-steady state). The list of experiments is shown in Table 3.

## 4    Data processing

### 4.1    Relationship between particle signal width and detected particle size voltage

To confirm the PSW variability, the accumulated relative PSW frequency is plotted for each field campaign in Fig. 6 when cloud water was detected based on a DOP threshold of 0.5. The approximate thickness of the cloud layer for each case is listed in
Table 1. All the CPS profiles contain PSWs smaller than 1.0 ms with 60%–80% relative frequency. There are three hypotheses





to account for this small PSW: (1) the detected thickness of the cross-section is thinner than 0.5 cm; (2) the flow speed in the CPS inlet is faster than the ascending speed (typically $5\pm1$ m s$^{-1}$); and (3) a combination of the above two hypotheses.

The first hypothesis is verified in section 2 (Fig. 3). Only 30% of the cross-section of the inlet (magenta area in Fig. 3b) can be used for a detection area with a 0.5-cm thickness, while nearly 70% of the area (other colors in Fig. 3b) can be used for a
thinner detection zone. It is therefore natural that the smaller PSW was frequently observed in each CPS flight.

We next introduce the possibility of variable flow speeds in the CPS inlet to verify the second hypothesis. Here, we focus on two cases: one with a standard slop curve (NY20-CPS03) and another with a steep slope curve in the smaller PSW range (NY20-CPS09). The mean ascending speeds where liquid clouds were detected was 5.0 and 6.1 m s$^{-1}$, respectively (Table 1). Faster vertical speed might contribute to the steep slope of the PSW frequency. The estimated PSW weighted by the detection
thickness in the CPS inlet (0.5 cm: 30%, 0.25 cm: 70%) is 0.65 and 0.53 ms for each case. The difference in the ascending speed therefore has a partial impact on the PSW variability.

Here, we consider the idealized flow distributions in the CPS inlet. The first case is assumed to have a relatively low Reynolds number ($Re$) with a slow flow speed distribution (relatively laminar flow). The second case is taken to have a relatively high $Re$ with a fast flow speed distribution (relatively turbulent flow). Once the flow distribution is given, the time to pass the cross-
section area is calculated (Fig. 7). The low flow speed case allows the large PSW ($>1.0$ ms), while the high flow speed case allows the small PSW ($<0.4$ ms). Another important finding in this calculation is that a significantly high-PSW area can exist near the CPS inlet wall owing to the low flow speed. The frequency slope for both cases is reproduced in some sense compared with real cases (Fig. 6d). The observed accumulative relative frequency of PSW would therefore be caused by this non-uniform flow in the CPS inlet and PSW distributions rather than the effect of particle overlapping, as proposed by Fujiwara et al. (2016).
Based on the verification of these two hypotheses, the third hypothesis is also correct. The combination of a non-uniform detection depth and flow distribution can explain the small PSW value.

### 4.2 Relationship between mean PSW and Reynolds number

There is no way to observe the flow speed in the CPS inlet during a normal CPS flight; however, using the observed parameters, the environment characteristics of the flow speed can be estimated. The idealized situations (Fig. 7) are first investigated
(assuming cases NY20-CPS03 and NY20-CPS09). A bulk flow speed is defined as $v_b = (0.5 \times frac + 0.25 \times (1 - frac)) \times 10^{-2}/(\overline{<PSW>} \times 10^{-3})$ (unit: m s$^{-1}$) to better understand the mean state in the CPS inlet, where $frac$ is the fraction of the detection area with 0.5 cm, and $\overline{<PSW>}$ is the spatially and temporally averaged PSW in the detection domain. The $v_b$ for low and high flow speed cases is estimated as 3.3 (NY20-CPS03) and 4.7 m s$^{-1}$ (NY20-CPS09), respectively. These values are roughly similar to the mean flow speed of 3.4 (Fig. 7a) and 5.5 m s$^{-1}$ (Fig. 7b) for each case simply because the estimated
mean PSWs (0.95 and 0.65 ms in low- and high-flow cases, respectively) are very similar to the observed values (0.99 ms in NY20-CPS03, 0.69 ms in NY20-CPS09) (Table 1). The $v_b$ would therefore be a potential indicator as a first approximation of the flow speed in the CPS inlet.

The observed $\overline{<PSW>}$ and $v_b$ are then calculated by time-averaging the observed PSWs in the liquid cloud layer (Table 1). The $v_b$ is generally weaker than the mean ascending speed ($v$). The question then arises of what controls $v_b$. Based on the





results presented in the previous subsection, $Re$ is a suitable indicator to characterize the flow distribution in the CPS inlet.

$$Re = \frac{v_b l}{\nu} = \frac{\rho l v_b}{\mu} = \frac{\rho l v_b}{\mu_0 \left(\frac{T}{T_0}\right)^{\frac{3}{2}} \cdot \frac{T_0 + S}{T + S}}, \tag{3}$$

where $l$ (m) is the scale of the inlet (= 0.01 m), $\nu$ (m$^2$ s$^{-1}$) is the kinematic viscosity coefficient, $\rho$ is the air mass density (kg m$^{-3}$), $\mu$ and $\mu_0$ are dynamic viscosity coefficients (Pa·s) under temperatures $T$ and $T_0$ (= 288.15 K), respectively, and $S$ is the Sutherland's constant (= 119.4 K).

By definition, $Re$ decreases with increasing $\nu$. This situation can occur under lower pressure because $\rho$ deceases, which suggests that there is a relationship between $\nu$ (= $\mu/\rho$) and $v_b$. The correlation coefficient between the two parameters among all the CPS sondes (except for tethered balloons) is relatively high (0.51, p-value: 0.015). The $Re$ in NY20-CPS09 (3332) is larger than in NY20-CPS03 (2680), which suggests that a more mixed flow regime is expected in the former. The CPS inlet flow characteristics therefore depend on air density (i.e., air pressure) as well as the vertical ascending speed, as discussed in

section 6.1; however, $v_b$ might be more complicatedly influenced by other factors, as discussed in section 6.2.

### 4.3 Cut-off PSW to reduce the unrealistic data

As shown in Fig. 7, a high PSW (e.g., >2 ms) is observed near the wall of the CPS inlet owing to the slow flow speed. The height where particles are detected under lower flow speed situations thus does not represent the observed height but rather the lower height owing to the time lag. Furthermore, the fact that a higher voltage of $I_{55}$ is observed with the higher PSW

(discussed later in Fig. 12b) suggests that the threshold PSW value can be useful to reduce unrealistic particle size data. This procedure is thus critical to estimate the effective cloud particle radius.

The PSW cut-off value (hereafter, $PSW_c$) is proposed as follows.

$$\overline{PSW_c} = \overline{PSW_{max} - PSW_{min}}, \tag{4}$$

where $PSW_c$ is the difference between the maximum and minimum PSWs ($PSW_{max}$ and $PSW_{min}$) counted per unit time

(= 1 s). Note that the number of data is six at most second readings. This procedure thus excludes at least one datum among the six recorded values per second. The overbar indicates the time average where a liquid cloud is observed (typically 50–100 s depending on the cloud-layer thickness). If the PSW is recorded randomly in a detection domain, the ratio of rejected data would be approximately 17% and the effective data ratio would be approximately 83%.

As a trial, the relationship between $\overline{PSW_c}$ and accumulative relative PSW frequency is investigated using the idealized cases

described in the previous subsection. The sampling points in the idealized CPS inlet (1001×1001 grids: 0.01 mm resolution) are randomly selected (black dots in Fig. 7). The $\overline{PSW_c}$ is calculated using 80 sets of six PSW data, which assumes 80-s observations in the cloud layer (a roughly 400-m-thick cloud). The estimated $\overline{PSW_c}$, shown in Fig. 7 as white contours, heavily depends on the flow speed distribution. The frequency up to $\overline{PSW_c}$ corresponds to 85% and 94% (triangles in Fig. 6d), which suggests that this data screening would be effective despite the limitation of the number of data if the point selection

is randomly determined with some time averaging. Using the real cases, $\overline{PSW_c}$ also ranges from 80% to 90% (triangles in





Fig. 6a–c: except for NY20-CPS01 owing to the thinner sampling depth of 100 m), which supports the randomness of particle counts in the CPS inlet.

## 4.4 Estimation of effective particle size

Fujiwara et al. (2016) conducted laboratory experiments to measure the particle voltage ($I_{55}$: V) for known particle diameter

sizes ($d$: μm) between 1 and 100 μm. Table 2 summarizes their results. However, they did not provide an empirical equation for estimating particle size and some approximations are required to estimate LWC and LWP. Based on the quadratic regression between $log_{10}(d)$ and $log_{10}(I_{55})$ (correlation coefficient = 0.983, p-value: 7.28 $\times 10^{-5}$), the following empirical equation is proposed:

$$log_{10}(d) = \sqrt{\frac{log_{10}(I_{55}) + 0.13303}{0.4257}} + 0.09831, \tag{5}$$

where $d$ is the diameter that corresponds to the observed voltage ($I_{55}$) of a particle. Considering that the number of $I_{55}$ data is six per second at most and one of which will be excluded where $PSW_{max}$ is larger than $PSW_c$, only five data sets are available to estimate the particles sizes. Although random sampling is assumed, as discussed in section 4.3, time-averaging in a certain thickness would better represent the section of cloud layers. Here, the averaging time is set to $\pm 2$ s, corresponding to a 25-m-thick cloud layer. The effective radius ($r_e$: μm) of the cloud droplets is estimated by considering volume averaging as

follows:

$$r_e = \frac{\left(\sum_{k=1}^{n} d_n^3\right)^{1/3}}{2n}, \tag{6}$$

where $n$ is the number of observations in the target cloud layer (typically 5 particles s$^{-1}\times$ 5 s = 25 particles) and $d_n$ is the $n$th particle diameter estimated by Eq. (5).

## 4.5 Correction factor of total particle count

Once the $r_e$ and total particle count are determined at each level, the LWC and LWP can be estimated as the integrated LWC. Although the observed total particle count should be corrected owing to sampling overlap, as proposed by Fujiwara et al. (2016), the sampling overlap might be a minor factor because most PSW are smaller than 1.0 ms with 70% of the accumulative relative frequency. Instead of this issue, sampling loss should be considered as a major factor because the CPS inlet flow dynamics during ascent have large uncertainties. In particular, there is no conclusive evidence that the ascending speed equals

the CPS inlet flow speed.

Here we consider the shape of the CPS housing with a radiosonde. The flow at the top of the housing during ascent (assuming 5 m s$^{-1}$) would be modified and aerodynamically slowed (<5 m s$^{-1}$). This reduced inflow into the CPS inlet leads to a loss





of cloud particle counts. Furthermore, the bulk flow speed $v_b$ in the CPS inlet estimated by $\overline{<PSW>}$ is generally slower than the ascending speed $v$ (Table 1). Fujiwara et al. (2016) stated that the flow speed at the bottom of the inlet (i.e., bottom

side of the CPS housing) is the same as the ascending speed. However, the downstream flow is heavily modified by the CPS cubic-shaped housing, thus causing a large pressure drag with turbulent wakes near the bottom side of the CPS housing. The CPS housing's background flow and flow features in the CPS inlet should therefore be carefully considered to correct the cloud particle count.

 For this purpose, we calculated the flow pattern around the CPS housing using Flowsquare$^+$. Fig. 8 shows the flow pattern

assuming an ascending speed of 4, 5, and 6 m s$^{-1}$. As expected, the flow speed is reduced at the top of the housing for each case. The minimum flow speed at the entrance of the CPS inlet is 0.69, 0.87, and 1.04 m s$^{-1}$ for each case, which is 17.2%±0.04% of the initial flow speed (Fig. 9a), which suggests that the chance that the air mass can enter the CPS inlet in a unit of time is reduced to 17.2%. The remaining air mass turns aside from the CPS housing by the divergent flow (Fig. 8). The correction factor for total particle counts is therefore proposed as 5.8 ($= 1/0.172$). Under a minimal ascending speed (exp-1m)

and assuming the tethered balloon measurements (green line in Fig. 9a), the reduction ratio is 16.7% (i.e., correction factor of 6.0).

 An additional remarkable feature is that the flow speed in the CPS inlet is recovered to the ascending speed with approximately 10% loss (Fig. 9a). Figs. 8 and 9 indicate that the flow speed in the CPS inlet increases with increasing ascending speed. Compared with the three cases, the pressure gradient between the top and bottom sides (i.e., pressure drag) regulates the flow

speed in the CPS inlet (Fig. 9b). The flow speed near the bottom side decreases, which is consistent with the results of Fujiwara et al. (2016) in terms of a slight decrease in flow speed compared with the CPS ascending speed; however, the mechanism differs.

## 5 Comparison with other data sources

### 5.1 Total particle count by a tethered balloon in the Arctic Ocean

The OPC's vertical profiles on the tethered balloon are used to evaluate the corrected total particle count by the CPS sonde. The OPC's count is based on a 5-s suction (L$^{-1}$), whereas the CPS's count is based on a 1-s interval. Thus, the CPS count is averaged by 5 s. The data in which the DOP values are larger than 0.5 are used for comparison, focusing on the liquid cloud. The CPS count unit (s$^{-1}$) is standardized to that of the OPC (L$^{-1}$) by the ascending speed at each level (typically ~1 m s$^{-1}$) and cross-sectional area of the CPS inlet (1 cm$^2$). Based on the extra simulation assuming an ascending speed of 1 m s$^{-1}$, the

correction factor of the CPS total count of 6.0 is applied to this case.

 Fig. 10 shows the vertical distribution of the number concentration of particles larger than 2 $\mu$m obtained by the OPC and CPS sonde. A 50-m-thick cloud layer characterizes case 1 at 400 m height where the OPC detected a peak value of around 10,000 L$^{-1}$, whereas the CPS significantly underestimates this value. This discrepancy arises from the low cloud cover of the thin stratus clouds (Fig. 5a), introducing horizontal and vertical heterogeneity in the measurements because of the vertical

distance between both systems of 5 m with a slight tilting. Sunlight (Fig. 5a) might affect to count of the signal (the DOP



values lower than 0.5 under high air temperature do not indicate ice cloud particles' existence). Nevertheless, the top and bottom of the cloud are consistent with values of approximately 2000–3000 $L^{-1}$. The second case is the thickest cloud case among the three (Fig. 5b). The observation terminates at 750 m height, but the moist layer (relative humidity $> 95\%$) continues until approximately 1200 m, as confirmed by a regular-time Vaisala RS-41 radiosonde observation (not shown). A cloud

bottom height of 600 m with 97% relative humidity matches well where the number concentration starts to increase. The rapid increase in concentration where the relative humidity is 100% is also very similar. Although both sensors detect a lower number concentration up to 700 m height, the remarkable difference between the two occurs at heights between 700 and 750 m. The OPC value continuously decreases, whereas the CPS value rapidly increases. This discrepancy arises from the detectable range of the sensors because the CPS has a wider particle size range, which suggests that larger cloud particles dominate at this level.

The averaged $I_{55}$ voltage at 600–650 and 700–750 m corrected by $\overline{PSW_c}$ is 0.61 V and 1.27 V, respectively, which corresponds to $\sim 2$ and $\sim 25$ $\mu$m in diameter (Table 2). The former particles are detected by the OPC, whereas the latter are likely out of range. The third case is the intermediate case in terms of the cloud layer (120 m thick) (Fig. 5c). The two concentration peaks at heights of 690 and 730 m are well-matched despite the CPS underestimation. The third peak in the CPS sonde at 760 m height is characterized by the larger particles out of the OPC range.

The total particle count corrected by the factor proposed by Fujiwara et al. (2016) is nearly the same as the raw particle count, leading to a significant underestimation (green line in Fig. 10). Because the slow ascending speed promotes $5/v$ larger than the PSW, the factor is frequently unity (i.e., 1.0) by the definition of Fujiwara's factor. A correction factor of 6.0 proposed in this study assumes that the ascending speed is 1.0 m s$^{-1}$; however, the speed is typically decreasing (e.g., 0.1–0.5 m s$^{-1}$) at the beginning of the launch and before reaching the maximum height owing to operational procedures. The correction factor may

therefore be larger than 6.0, which would be in better agreement with the OPC results. Despite the difference in the detectable particle size range and sampling method (suction vs. natural ventilation by the ascending motion) between the OPC and CPS sonde, the correction factor for the CPS's total particle count proposed in this study offers a promising advantage to provide meaningful physical information for the quantitative analysis of cloud microphysics processes.

## 5.2   LWC and LWP by microwave radiometry at Ny–Ålesund, Svalbard

Using the land-based remote sensing product at Ny–Ålesund, $r_e$ and $PSW_c$ are applied to estimate and validate LWC and LWP. The Cloudnet product is only available for a portion of the March 2020 data set to retrieve the LWC. Only the NY20-CPS03 case is available for comparison. Because this case is the single-layer cloud case (400 m water cloud depth) and the LWP from the Cloudnet product is 30.4 g kg$^{-1}$ at 17:10 UTC on the measurement day, which is larger than the typical uncertainty of the HATPRO (20–25 g m$^{-2}$), comparison with the CPS sonde is feasible. Although two more cases might be available

(NY20-CPS09 and NY20-CPS14), they are multiple-layered clouds with thinner water cloud layers of less than 200 m depth. As seen in the previous subsection, the CPS sonde tends to underestimate the total particle count, presumably causing an underestimation of LWC and LWP. Additionally, the LWP observed by HATPRO on those days is 12.0 and 3.4 g kg$^{-1}$, which suggests that these observed values also contain considerable uncertainty.





To estimate LWC, $r_e$ is calculated at each level by satisfying the $PSW_c$ threshold value to Eqs. 5 and 6. The LWC can be
estimated assuming the cloud droplet shape is a sphere and water density is 1000 kg m$^{-3}$. Fig. 11 shows the vertical profiles of
air temperature, relative humidity, $I_{55}$ voltage, DOP value, total particle count, $PSW_c$, $r_e$, and LWC. This case is characterized
by mixed-phase clouds where the lower layer up to 500 m is filled by cloud ice or snow (i.e., the DOP value is small; blue dots
in Fig. 11), whereas the upper layer from 500 to 900 m is dominated by cloud water (e.g., the DOP is larger than 0.5; red dots
in Fig. 11).

Based on the vertical distribution of the $I_{55}$ voltage, $r_e$ increases from ∼10 to 25 $\mu$m with two peaks at 700 and 830 m.
$PSW_c$ ranges between 1 and 5. The $I_{55}$ voltage sometimes exceeds 7 V, which suggests that $PSW_c$ appropriately reduces the
samples larger than the CPS detection limit or solid cloud phase. The LWC increases up to 850 m with a maximum of 0.25
g m$^{-3}$. This peak value does not depend on the total particle count but rather the size of $r_e$. These characteristics generally
agree with the adiabatically retrieved LWC by the Cloudnet product, which linearly increased up to the cloud top. The vertical
integration of LWC, namely LWP, shows that the CPS sondes (21.6 g m$^{-2}$) tend to underestimate the LWP by the Cloudnet
(30.4 g m$^{-2}$). A possible reason might be cloud ice contamination. The DOP threshold between cloud ice and cloud water is
set to 0.5 in this study. If a more strict DOP threshold is applied, the LWP increases (e.g., to 27.9 g m$^{-2}$ for a DOP threshold
of 0.7); however, the number of samples for the $r_e$ calculation decreases with considerably higher uncertainty of the LWC
calculation. It should also be noted that the LWP data from the Cloudnet product also have a given uncertainty, as previously
mentioned before. In other words, the LWP value by the CPS sonde falls within the range of the Cloudnet product uncertainty.

If the correction factor by Fujiwara et al. (2016) is applied to this case, the total particle count (∼$10^4$ L$^{-1}$: green dots in Fig.
11) is one order larger than our corrected value (∼$10^3$ L$^{-1}$), and thus overestimates LWP (506 g m$^{-2}$). Although the true $r_e$ is
unknown, a combination of corrected total particle count and $r_e$ using $PSW_c$ can provide new insight into understanding the
vertical structure of liquid phase clouds.

## 6   Discussion

### 6.1   Limitation for estimating LWP by CPS sondes

Because ERA5 (Hersbach et al., 2020) can qualitatively simulate liquid-phase clouds in the lower troposphere over the ice-free
ocean (Inoue et al., 2020), a comparison of CPS-derived LWP ($LWP_{CPS}$) with ERA5 ($LWP_{ERA5}$) can provide additional
insight on the estimation of $LWP_{CPS}$, in particular how unrealistic an obtained value may be. A comparison with the hourly
ERA5 outputs at the closest grid point of the ship is made in the case of the Arctic cruises from 2018 and 2019 to avoid
topographic effects at Ny–Ålesund in ERA5. Fig. 12a shows a scatter plot between $LWP_{ERA5}$ and $LWP_{CPS}$. Several outliers
show a common feature: a mean $r_e$ larger than 20 $\mu$m (gray dots). By excluding these five cases, the correlation coefficient
between $LPW_{ERA5}$ and $LWP_{CPS}$ is 0.55, with a p-value of 0.082. $LWP_{CPS}$ is almost twice as much as $LWP_{ERA5}$
because ERA5 uses a coarser vertical resolution (seven layers below 850 hPa). The cloud microphysics without solving each
hydrometeor number concentration would be the other factor. Of course, several error sources can arise from the corrected CPS





sonde data. In any case, abnormal $LWP_{CPS}$ values would occur in the case of relatively large particle sizes, which are larger in $I_{55}$.

The MR19-CPS06 case (largest $r_e$ case: 31 $\mu$m) reveals that the voltage in $I_{55}$ frequently reaches the maximum regardless of the degree of PSW (red circles in Fig. 12b), whereas the MR19-CPS07 case (normal $r_e$ case: 14 $\mu$m) does not exhibit such a

condition (blue squares in Fig. 12b). The former case has a larger $PSW_c$ of 2.75 ms (red dashed line Fig. 12b), which cannot correctly exclude the saturated voltage data and thus causes unrealistic $r_e$ and $LWP_{CPS}$. The latter case successfully leaves the data via $PSW_c$ (blue dashed line in Fig. 12b). In the intermediate $PSW_c$ case with 2.07 ms, the number of saturated $I_{55}$ voltage is reduced (MR19-CPS09: green dashed line in Fig. 12b); however, the $LWP_{CPS}$ is still seven times larger than $LWP_{ERA5}$ with $r_e$ = 22 $\mu$m. Extra caution is therefore needed for high $PSW_c$ (e.g., >2.0 ms).

## 6.2 Other sources to modify the CPS inlet flow speed

The simulations show that the flow speed in the CPS inlet becomes fast with increasing ascending speed ($v$) (Fig. 8), leading to a decrease in $\overline{PSW_c}$ under relatively high $Re$ conditions (turbulent flow). The correlation coefficient between $v$ and $\overline{PSW_c}$ is −0.58 (p-value: 0.0023) if the tethered balloon cases are included. The pressure height ($p$) is another factor to modify $\overline{PSW_c}$ (correlation coefficient = 0.57, p-value: 0.0027). The multiple linear regression correlation coefficient to predict $\overline{PSW_c}$ with

$v$ and $p$ is 0.71 with an F-value of 0.0003. Therefore, both $v$ and $p$ are important environmental parameters for determining $\overline{PSW_c}$.

Fujiwara et al. (2016) monitored the CPS inlet flow speed by attaching a duct with anemometers at the bottom of the CPS inlet. They concluded that the CPS inlet flow speed is nearly the same as the ascending speed (Fig. B1 in Fujiwara et al. (2016)); however, the difference between them increases with increasing height, particularly in the stratosphere. Here, the sensitivity of

the ambient air pressure against exp-5m under 850 hPa is also calculated by changing the pressure (925, 700, 500, 300, 100, and 50 hPa: Table 3). The air temperature is fixed as an idealized situation. As expected, the CPS inlet flow speed decreases with decreasing ambient air pressure, particularly above 300 hPa (blue line in Fig. 13), which agrees with the results of Fujiwara et al. (2016). Accordingly, the correction factor is increased from 5.8 to 6.6; however, for the investigation of clouds in the troposphere, air pressure sensitivity to the correction factor is small (i.e., the fixed value would be practical).

Although half of the variability of $\overline{PSW_c}$ can be explained by $v$ and $p$, the remainder may be related to other potential factors, including (1) tilting of the CPS housing induced by horizontal winds; (2) rotation of the CPS housing; and (3) swing between the CPS sonde and balloon (20 m distance). These effects might slow the flow speed near the inlet wall and increase the chance of cloud particles merging with each other. Because the pressure gradient between the top and bottom sides of the CPS housing controls the CPS inlet flow speed (Fig. 9), the impact of horizontal wind speed on the pressure fields should be

verified. Additional simulations were thus performed assuming horizontal winds ($v_h$) of 2.5 and 5 m s$^{-1}$ (Table 3) under the $v$ of 5 m s$^{-1}$ to understand how the wind angle against the CPS housing modifies the pressure field. As expected, the pressure and flow patterns differ substantially from the experiments without horizontal wind (Fig. 14). In exp-h5m, the flow speed in the CPS inlet decreases compared with exp-5m even if the ascending speed is the same because the large pressure gradient is present at both lateral sides of the CPS housing rather than the top-bottom sides. Under actual conditions, the CPS housing





would be tilted by horizontal winds with rotation and swing, which complicates the relationships between horizontal wind speed and CPS inlet flow speed. The correlation between $v_h$ and $v_b$ is insignificant (0.33, p-value: 0.147).

## 7 Conclusions

The CPS sonde is a unique observation system to measure cloud phases and total particle counts and sizes; however, this system requires appropriate flight adapted corrections to obtain quantitative and meaningful results. Fig. 15 summarizes the procedure

to calculate LWP using the raw CPS data. In this study, a total particle count correction factor of 5.8 is proposed that considers the CPS housing shape. This value is smaller than that suggested by Fujiwara et al. (2016) under typical ascending speeds (5 m s$^{-1}$). A direct comparison with the OPC on tethered balloon measurements shows that this proposed correction factor can estimate the total particle count, particularly in the case of thick clouds (e.g., >100 m depth). The discrepancy between CPS sonde and OPC data occurs at the level where larger particles dominate (e.g., >10 $\mu$m), which is out of the OPC range.

In this study, we focus on a liquid-phase cloud in the first trial. To estimate LWC and LWP, an empirical formula to calculate the effective radius $r_e$ from the $I_{55}$ voltage is proposed based on laboratory experiments by Fujiwara et al. (2016). However, the $I_{55}$ voltage sometimes contains outliers close to the maximum CPS voltage limit. In such cases, the PSW value, which is the time interval of each particle signal, also increases (Fig. 12b). The PSW has previously been considered as a constant (1.0) if the flow speed is 5 m s$^{-1}$, assuming that the detectable length is 0.5 cm; however, the observable length varies from 0 to 0.5

cm owing to the photodetector angle. Therefore, nearly 70% of PSWs are usually smaller than 1.0 ms, as shown in Fig. 6. The next issue is thus how the remaining data (30%) are treated. According to a simplified situation that assumes the flow speed distribution in the CPS inlet, it is found that the large PSW is attributed to slower flow speeds near the wall of the CPS inlet (i.e., slower flow speeds require longer time intervals to pass the detection area). The reduced flow speed make the particles merge into a larger particle, which should be excluded from the estimation of $r_e$, LWC, and LWP. Although there are only six

samples of PSW per second, we propose a cutoff value of PSW ($PSW_c$), defined as the difference between the maximum and minimum PSW. The $PSW_c$ is not a constant but varies in each launch and second because PSW depends on the flow speed (ascending speed and possibly horizontal wind speed) and ambient air pressure. Although the validation is only in one case, the LWC and LWP estimated by the CPS sonde broadly capture the characteristics obtained by land-based remote sensing.

This study focused on the Arctic region from fall to spring, which favorably reduced the effects of sunlight for the CPS

sonde observations. However, additional nighttime field experiments at lower latitudes, at which the amount of moisture and clouds is larger than in the polar region would advance the evaluation of data from CPS sondes.

*Data availability.* The CPS sonde data are available upon request to the first author.



*Author contributions.*   JI, KS, and FT participated in the R/V Mirai cruises and carried out CPS sonde and tethered balloon observations. KS and YT engaged in the CPS observations at Ny-Ålesund. MM arranged the operation at AWIPEV station for the CPS sonde observations. JI

mainly analyzed the data and prepared the manuscript with contributions from all co-authors.

*Competing interests.*   The authors declare that they have no conflict of interest.

*Acknowledgements.*   This work was supported by JSPS KAKENHI (grant numbers 18H03745, 18KK0292, 19K14802) and NIPR general collaboration project no. 31-17. The activities in 2019 and 2020 were endorsed by the Multidisciplinary drifting Observatory for the Study of Arctic Climate (MOSAiC) project. We are greatly indebted to the officers and crew of RV Mirai. Onsite support by Junji Mat-

sushita and the AWIPEV observatory at Ny-Ålesund was very helpful. We thank Esther Posner, PhD, from Edanz Group (https://en-author-services.edanzgroup.com/ac) for editing a draft of this manuscript.



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





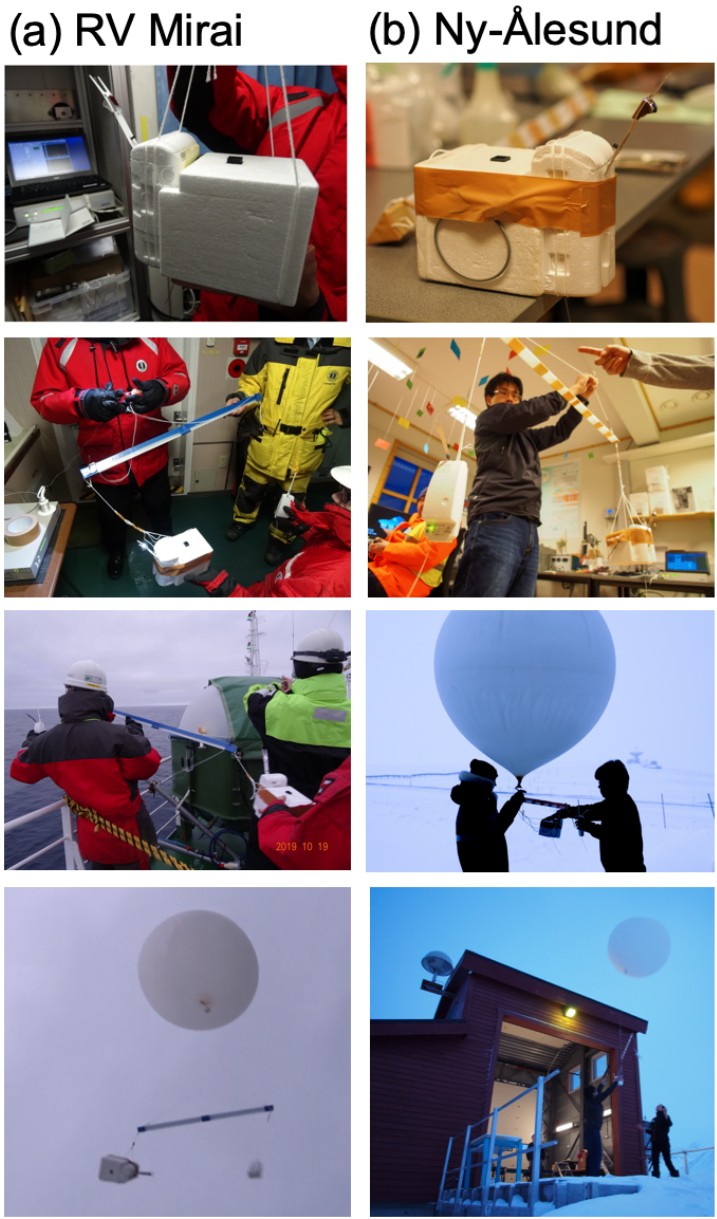

**Figure 1.** Photographs of the CPS sonde operation at the (a) RV Mirai in October 2019 and (b) Ny-Ålesund in March 2020. The CPS housing within a black inlet duct on top is connected to the Meisei RS-11G radiosonde. During the 2019 and 2020 campaigns, the Vaisala RS41-SGP was attached to the opposite side of the rod.



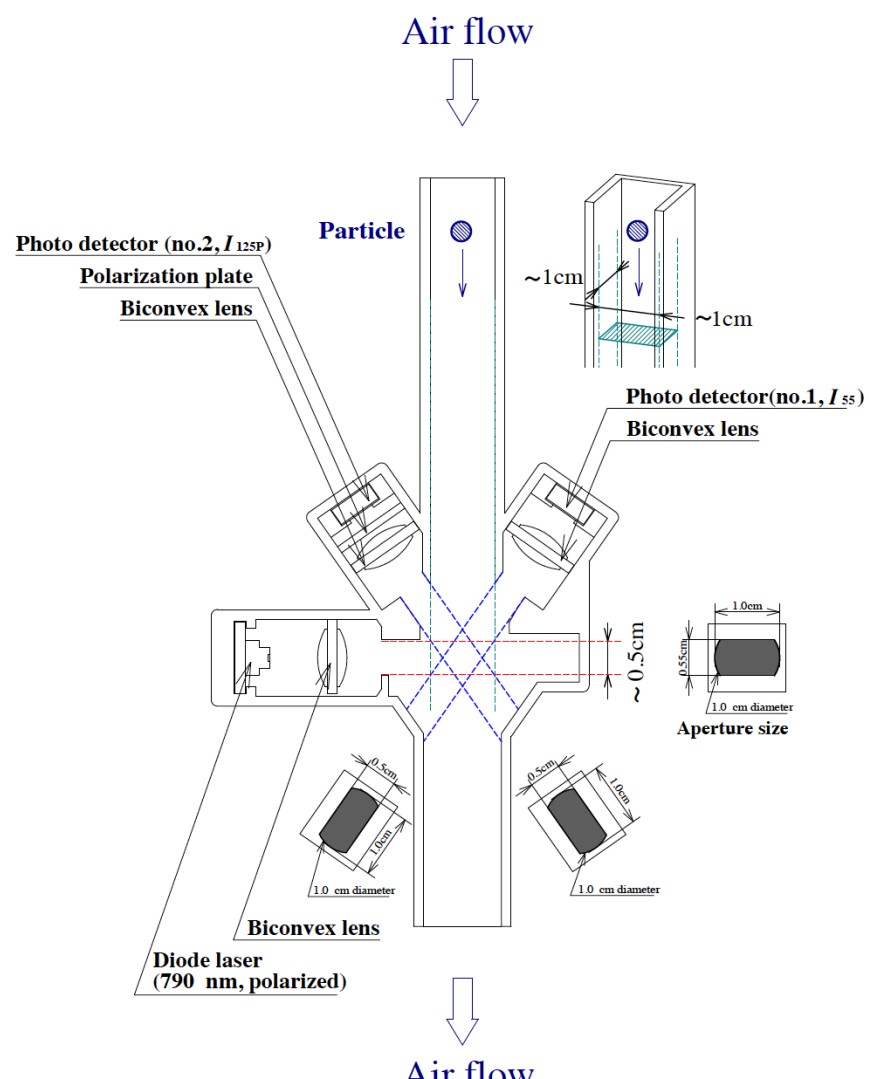

**Figure 2.** Schematic diagram of the CPS (from Fujiwara et al. (2016)).

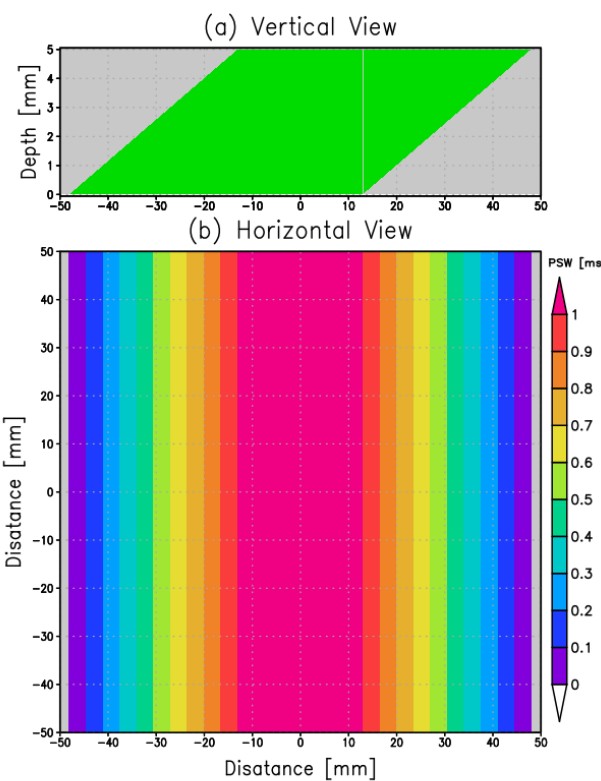

**Figure 3.** (a) Vertical CPS detection domain of the $I_{55}$ detector (green area) and (b) particle signal width (PSW) in the case of 5 m s$^{-1}$ flow speed (typical ascending speed of the CPS sonde).

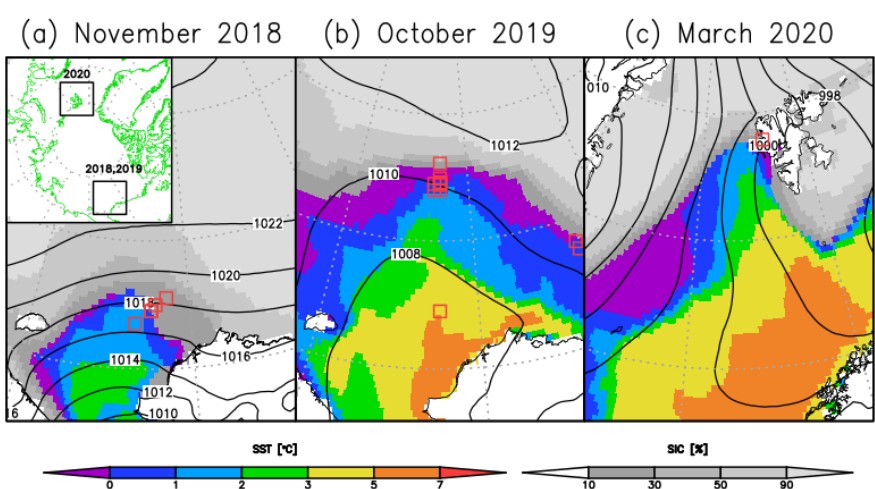

**Figure 4.** Location of the CPS sondes (red squares) during research cruises in (a) November 2018 and (b) October 2019, and (c) a field campaign in March 2020. Monthly mean sea-ice concentration (gray shading), sea surface temperature (color shading), and sea-level pressure (contours) are based on ERA5 reanalysis.



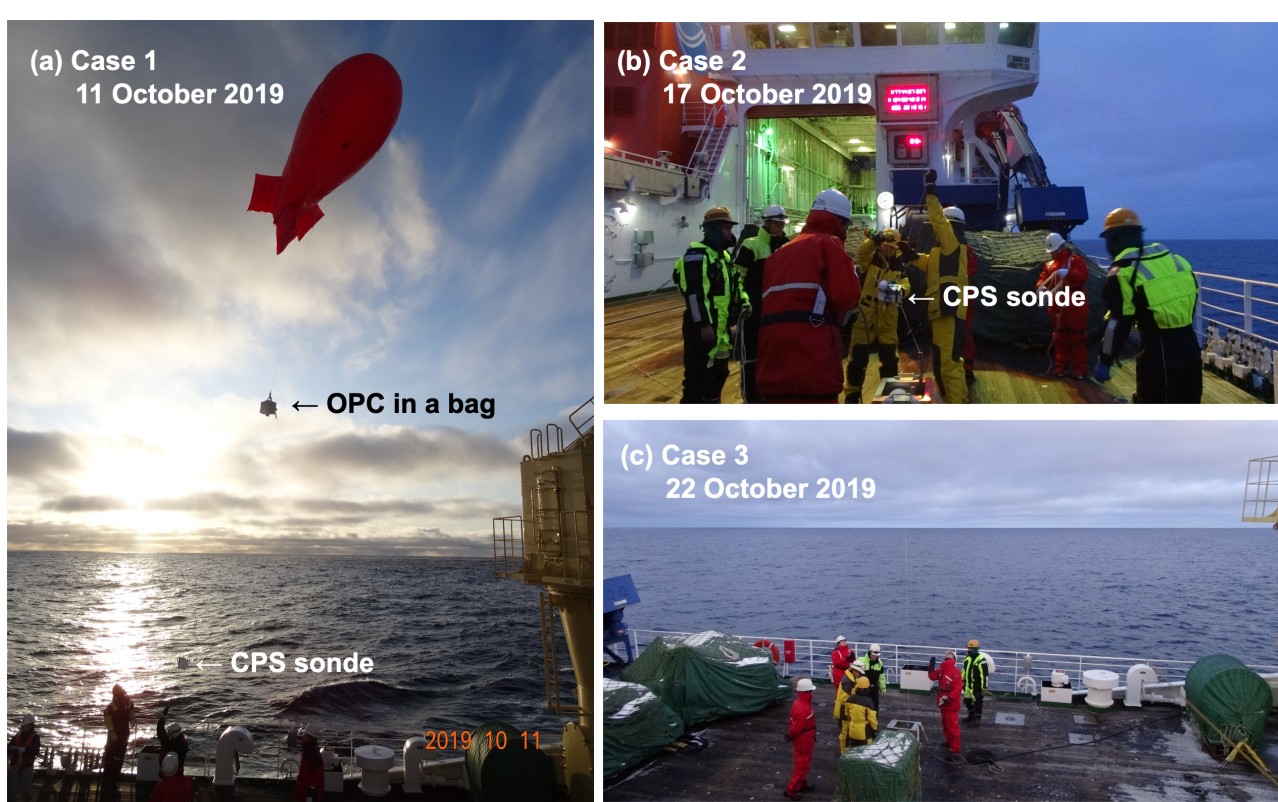

**Figure 5.** Photographs of tethered balloon measurements for the cases of (a) October 11, 2019 (MR19-CPST1), (b) October 17, 2019 (MR19-CPST2), and (c) October 22, 2019 (MR19-CPST3) on RV *Mirai* in the Arctic Ocean.



**Figure 6.** Accumulated relative frequency of PSW for (a) November 2018, (b) October 2019, (c) March 2020, and (d) idealized flow speeds (low/low Reynolds number case) compared with cases NY20-CPS03/09. Each triangle indicates the cut-off PSW (PSW$_c$).



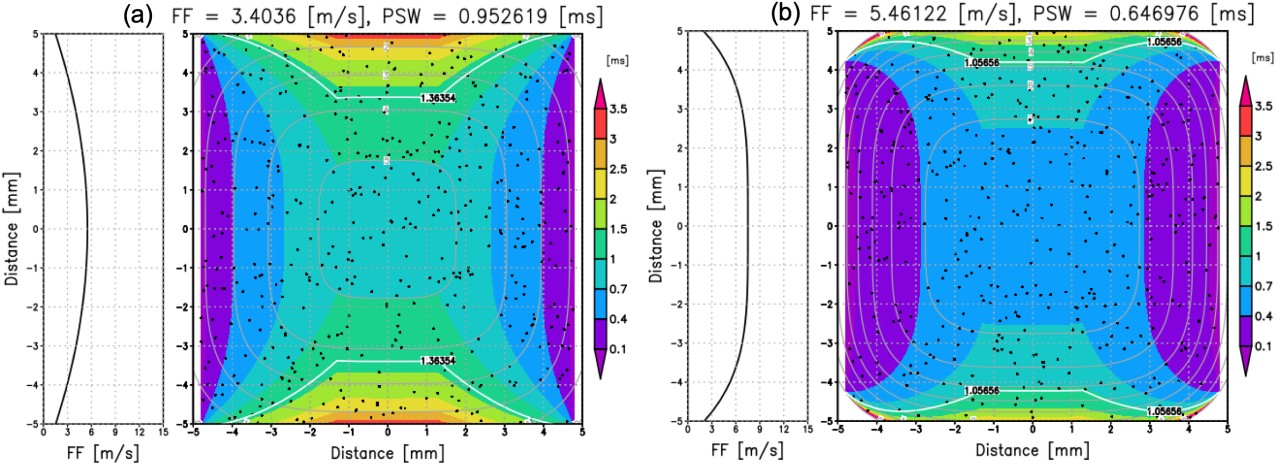

**Figure 7.** Idealized flow speed (gray contours) and PSW (colored shadings) distribution. (a) Laminar flow case. (b) Turbulent flow case. The spacial mean value of the flow and PSW are indicated at the top of the figure. White contours indicate the cut-off PSW (PSW$_c$). Black dots are the random sampling points assuming the 80-sec observation within a 400-m thick cloud layer.



**Figure 8.** Cross-section of the simulated flow speed around the CPS housing (absolute speed: m s$^{-1}$) assuming an ascending speed of (a) 4 m s$^{-1}$, (b) 5 m s$^{-1}$, and (c) 6 m s$^{-1}$. Contours and gray shades indicate the pressure difference from the initial state (Pa) and CPS housing.



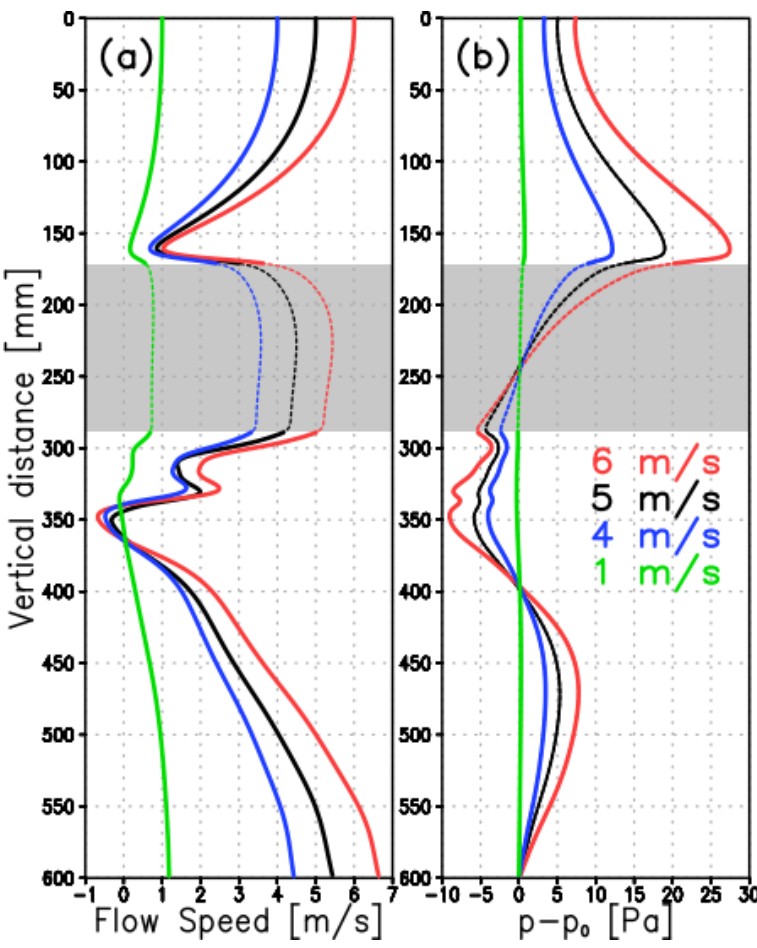

**Figure 9.** Vertical distribution across the CPS inlet of (a) vertical flow speed (m s$^{-1}$) and (b) pressure difference (Pa) from the initial state for each experiment (ascending speed is 1 m s$^{-1}$ (green), 4 m s$^{-1}$ (blue), 5 m s$^{-1}$ (black), and 6 m s$^{-1}$ (red)). Gray shading indicates the CPS inlet.



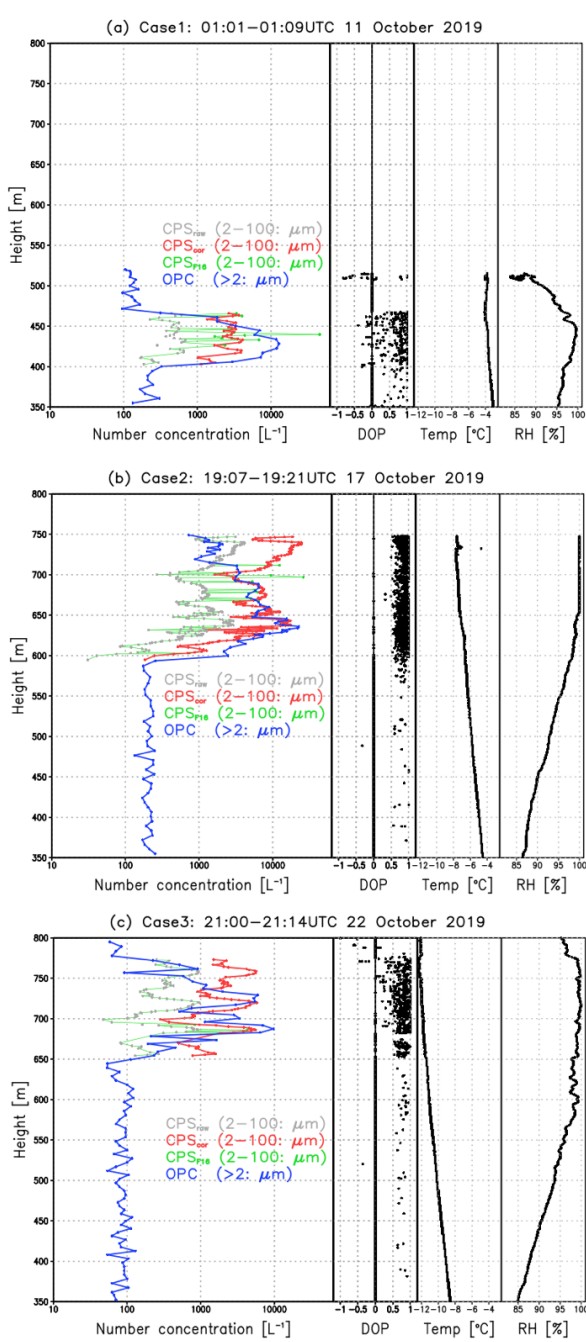

**Figure 10.** Vertical distribution of the number concentration of particles larger than 2 $\mu$m by the OPC (blue) and CPS sonde (corrected in this study in red; corrected by Fujiwara et al. (2016) in green) during the tethered balloon measurements on RV *Mirai*. Gray dots indicate the original CPS total counts. The value of DOP, air temperature, and relative humidity are indicated by black dots and black lines for each case.



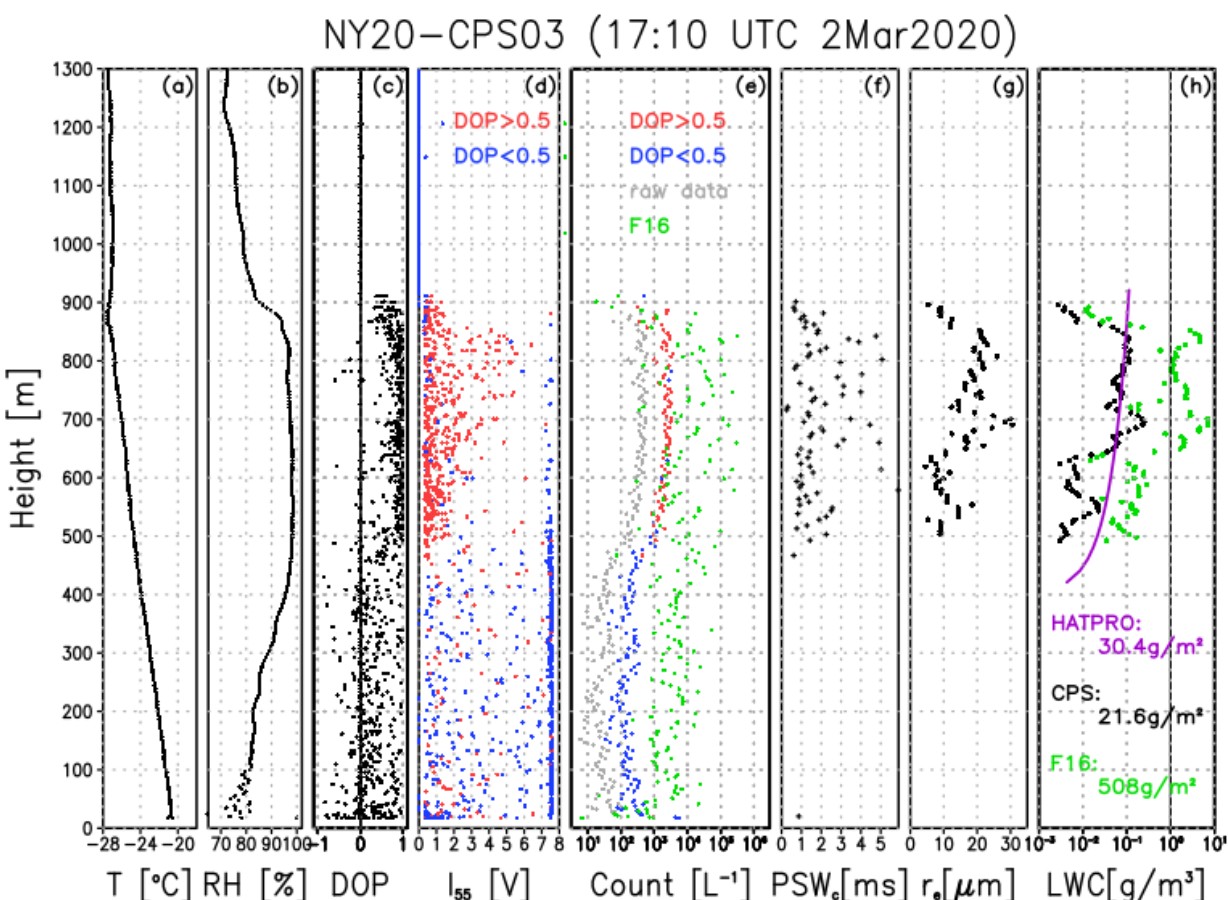

**Figure 11.** Vertical distributions of (a) air temperature (°C), (b) relative humidity (%), (c) DOP, (d) $I_{55}$ (V), (e) total particle count ($L^{-1}$), (f) $PSW_c$ ($s^{-1}$), (g) effective liquid particle radius, and (h) liquid water content (g m$^{-3}$). The numbers in (h) indicate the amount of the liquid water path (g m$^{-2}$) calculated by each method.

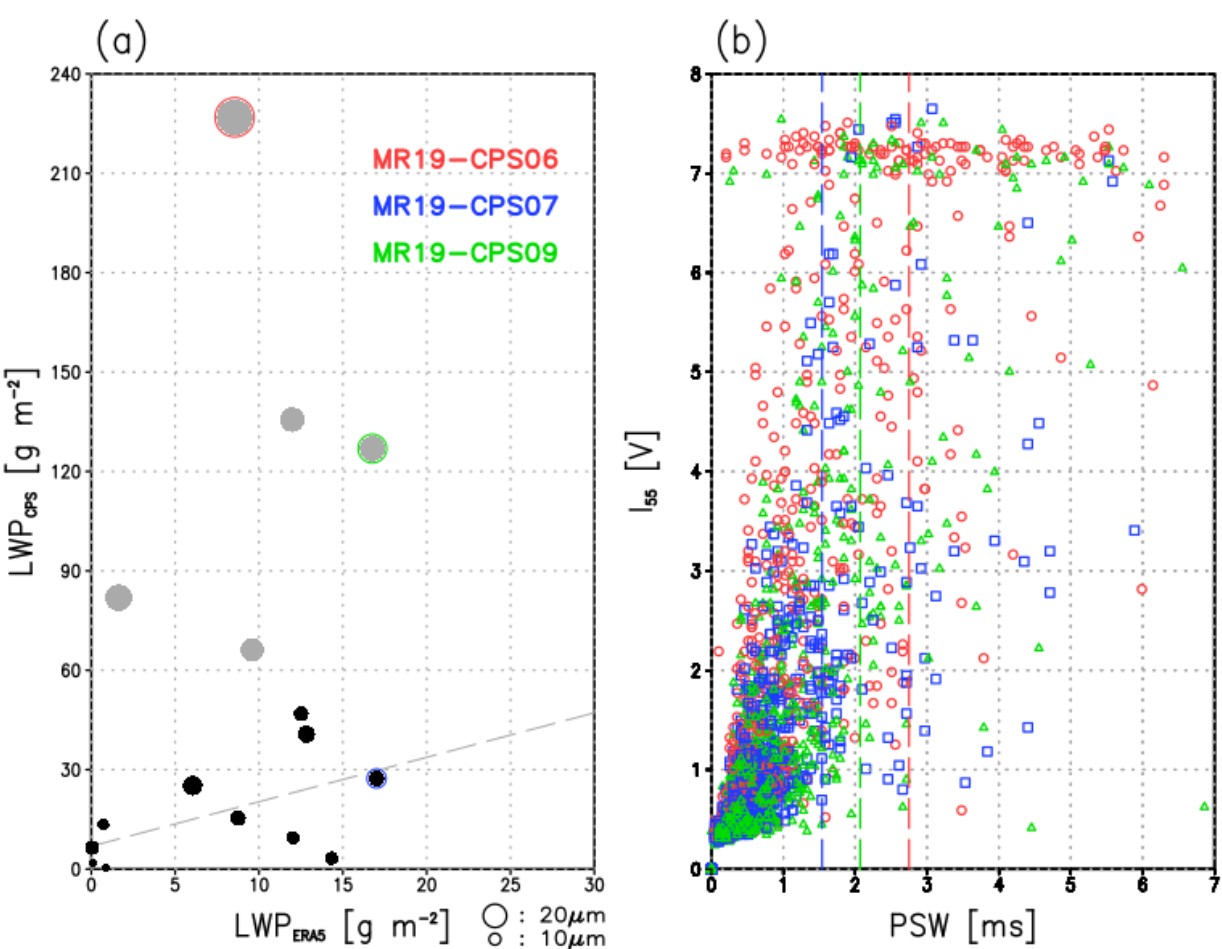

**Figure 12.** (a) Scatter plot of LWP (g m$^{-2}$) from ERA5 and CPS sonde for the cruises in 2018 and 2019 and (b) relationships between PSW (ms) and $I_{55}$ (V) for MR19-CPS06 (red), -CPS07 (blue) and -CPS09 (green). Dots in (a) indicate the relative size of the mean effective radius ($r_e$). Gray dots show the cases with $r_e$ larger than 20 $\mu$m. A gray dashed line indicates a linear regression line by excluding the cases with $r_e$ larger than 20 $\mu$m. Colored dashed lines in (b) show the cut-off PSW (PSW$_c$).





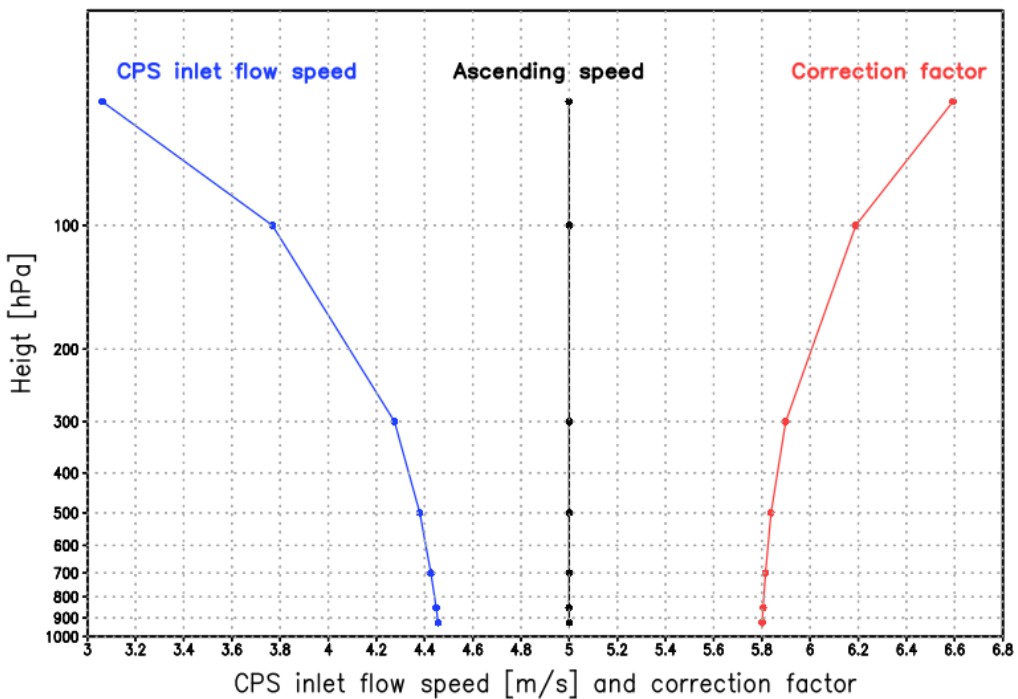

**Figure 13.** Vertical distributions of the modeled flow speed in the CPS inlet and correction factor.



**Figure 14.** Same as Fig. 8 but for different horizontal wind speeds: (a) 0 m s$^{-1}$; (b) 2.5 m s$^{-1}$; and (c) 5 m s$^{-1}$.



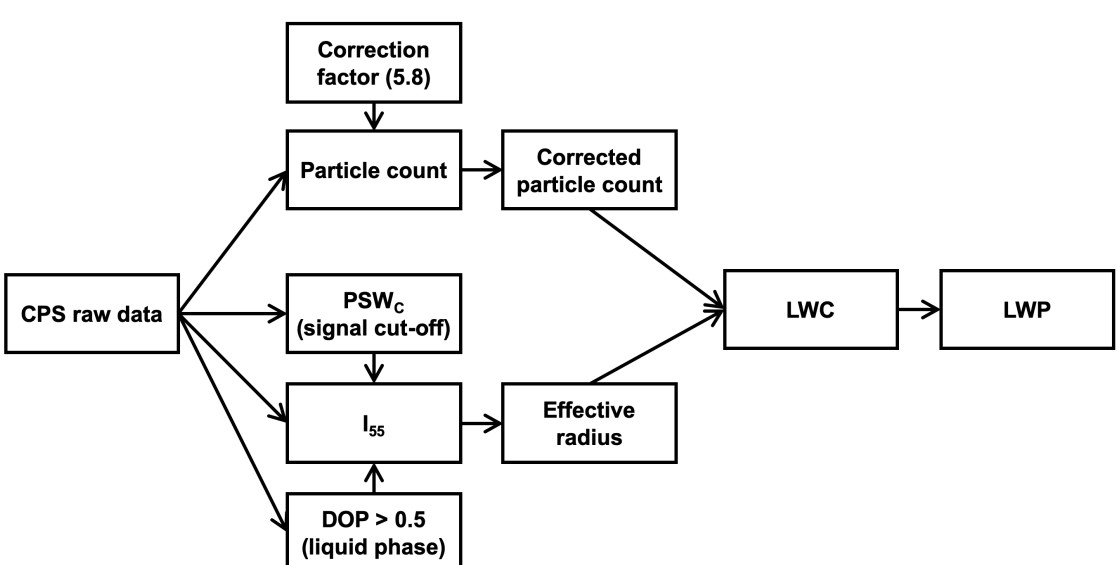

**Figure 15.** Flow chart for calculating LWP from the raw CPS data.



**Table 1.** Case number of the CPS launch, launch date, ascending speed, horizontal wind speed, air temperature, air pressure, height of cloud layer, air density, dynamic viscosity, kinematic viscosity, Reynolds number, mean PSW, cut-off PSW, bulk flow speed, and effective radius

| CPS no. | mm/dd/yy | hh:mm | $v$ | $v_h$ | $T$ | $p$ | height | $\rho$ | $\mu$ | $\nu$ | $Re$ | $\overline{<PSW>}$ | $\overline{PSW_c}$ | $v_b$ | $r_e$ |
|---|---|---|---|---|---|---|---|---|---|---|---|---|---|---|---|
| unit | - | UTC | m s$^{-1}$ | m s$^{-1}$ | °C | hPa | m | kg m$^{-3}$ | Pa·s | m$^2$ s$^{-1}$ | - | ms | ms | m s$^{-1}$ | $\mu$m |
| MR18-CPS05 | 11/13/18 | 12:04 | 5.3 | 6.7 | −15.1 | 873.3 | 1100-1500 | 1.18 | 1.64E-05 | 1.39E-05 | 2784 | 0.84 | 1.47 | 3.9 | 11 |
| MR18-CPS06 | 11/14/18 | 14:19 | 4.6 | 11.5 | −11.6 | 886.0 | 600-1650 | 1.18 | 1.66E-05 | 1.40E-05 | 2097 | 1.10 | 2.38 | 2.9 | 28 |
| MR18-CPS07 | 11/16/18 | 12:00 | 6.1 | 1.2 | −11.3 | 901.0 | 750-1250 | 1.20 | 1.66E-05 | 1.38E-05 | 2133 | 1.10 | 2.02 | 3.0 | 25 |
| MR18-CPS09 | 11/20/18 | 01:36 | 4.7 | 11.1 | −13.6 | 955.8 | 200- 650 | 1.28 | 1.65E-05 | 1.28E-05 | 2479 | 1.02 | 1.91 | 3.2 | 20 |
| MR18-CPS10 | 11/20/18 | 06:09 | 5.1 | 11.0 | −11.5 | 933.6 | 350- 850 | 1.24 | 1.66E-05 | 1.33E-05 | 3193 | 0.76 | 1.62 | 4.3 | 18 |
| MR18-CPS11 | 11/20/18 | 10:30 | 4.8 | 9.8 | −11.3 | 924.5 | 350-1000 | 1.23 | 1.66E-05 | 1.35E-05 | 2521 | 0.96 | 2.05 | 3.4 | 24 |
| MR19-CPS01 | 10/13/19 | 05:30 | 5.1 | 16.3 | −22.3 | 743.9 | 2200-2500 | 1.03 | 1.60E-05 | 1.55E-05 | 3180 | 0.66 | 1.12 | 4.9 | 6.9 |
| MR19-CPS02 | 10/14/19 | 05:29 | 5.9 | 11.6 | −13.6 | 834.7 | 1400-1600 | 1.12 | 1.65E-05 | 1.47E-05 | 3014 | 0.73 | 1.40 | 4.4 | 12 |
| MR19-CPS03 | 10/16/19 | 05:43 | 5.6 | 5.4 | −6.7 | 887.6 | 800-1200 | 1.16 | 1.68E-05 | 1.45E-05 | 2258 | 0.99 | 2.00 | 3.3 | 17 |
| MR19-CPS04 | 10/17/19 | 05:30 | 5.2 | 3.5 | −7.6 | 901.7 | 550-1200 | 1.18 | 1.68E-05 | 1.42E-05 | 2683 | 0.85 | 1.72 | 3.8 | 13 |
| MR19-CPS05 | 10/18/19 | 05:30 | 5.6 | 4.8 | −8.9 | 912.9 | 600-1000 | 1.20 | 1.67E-05 | 1.39E-05 | 3135 | 0.74 | 1.14 | 4.4 | 12 |
| MR19-CPS06 | 10/19/19 | 05:30 | 5.8 | 7.6 | −8.0 | 917.3 | 450-1100 | 1.21 | 1.68E-05 | 1.39E-05 | 1836 | 1.27 | 2.75 | 2.6 | 31 |
| MR19-CPS07 | 10/19/19 | 17:30 | 5.3 | 10.4 | −7.3 | 933.0 | 350-1000 | 1.22 | 1.68E-05 | 1.37E-05 | 2960 | 0.80 | 1.54 | 4.1 | 14 |
| MR19-CPS08 | 10/19/19 | 23:30 | 3.7 | 11.1 | −8.7 | 910.7 | 550-1200 | 1.20 | 1.67E-05 | 1.39E-05 | 2733 | 0.85 | 1.63 | 3.8 | 14 |
| MR19-CPS09 | 10/20/19 | 05:30 | 4.7 | 10.7 | −9.8 | 891.8 | 650-1450 | 1.17 | 1.67E-05 | 1.41E-05 | 2296 | 1.00 | 2.07 | 3.2 | 22 |
| MR19-CPS10 | 10/21/19 | 05:30 | 4.5 | 10.0 | −11.9 | 842.8 | 1000-1900 | 1.12 | 1.66E-05 | 1.47E-05 | 3373 | 0.65 | 1.07 | 5.0 | 7 |
| MR19-CPS11 | 10/22/19 | 05:30 | 4.4 | 2.9 | −15.0 | 829.8 | 1300-1800 | 1.12 | 1.64E-05 | 1.47E-05 | 2948 | 0.75 | 1.36 | 4.3 | 10 |
| MR19-CPST1* | 10/11/19 | 01:01 | 1.1 | – | −3.7 | 956.3 | 400- 470 | 1.37 | 1.70E-05 | 1.37E-05 | 1489 | 1.59 | 3.08 | 2.0 | 17 |
| MR19-CPST2* | 10/17/19 | 19:06 | 0.5 | – | −7.2 | 929.4 | 600- 750 | 1.38 | 1.68E-05 | 1.38E-05 | 1387 | 1.70 | 3.47 | 1.9 | 14 |
| ME19-CPST3* | 10/22/19 | 21:00 | 0.9 | – | −11.3 | 927.3 | 650- 780 | 1.34 | 1.66E-05 | 1.34E-05 | 2420 | 1.00 | 2.09 | 3.2 | 10 |
| NY20-CPS01 | 03/01/20 | 16:47 | 4.3 | 6.3 | −26.0 | 840.0 | 1350-1500 | 1.18 | 1.58E-05 | 1.34E-05 | 1878 | 1.29 | 1.04 | 2.5 | 7 |
| NY20-CPS03 | 03/02/20 | 17:02 | 5.0 | 8.2 | −26.1 | 915.5 | 500- 900 | 1.29 | 1.58E-05 | 1.23E-05 | 2680 | 0.99 | 1.83 | 3.3 | 15 |
| NY20-CPS07 | 03/04/20 | 04:49 | 4.5 | 12.3 | −16.1 | 828.5 | 1150-1950 | 1.12 | 1.64E-05 | 1.46E-05 | 2821 | 0.79 | 1.11 | 4.1 | 11 |
| NY20-CPS09 | 03/07/20 | 16:47 | 6.1 | 9.5 | −12.1 | 871.7 | 900-1150 | 1.16 | 1.66E-05 | 1.42E-05 | 3332 | 0.69 | 0.97 | 4.7 | 8 |
| NY20-CPS14 | 03/18/20 | 23:47 | 5.5 | 7.0 | −27.2 | 916.3 | 650- 800 | 1.30 | 1.58E-05 | 1.22E-05 | 2945 | 0.91 | 1.23 | 3.6 | 9 |

* A CPS sonde by the tethered balloon



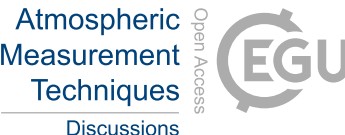

**Table 2.** Laboratory experiments to measure the CPS voltage for various standard particle sizes by Fujiwara et al. (2016).

| Diameter of standard particles | diameters for water | $I_{55}$ voltage |
|---|---|---|
| ($\mu$m) | ($\mu$m) | (V) |
| 1 | 1.36 | not sensitive |
| 2 | 2.10 | 0.648±0.538 |
| 5 | 5.93 | 0.791±0.838 |
| 10 | 13.25 | 0.717±0.557 |
| 20 | 26.65 | 1.33±1.24 |
| 30 | 39.50 | 2.16±1.82 |
| 60 | 79.78 | 5.36±2.68 |
| 100 | 132.87 | 6.66±1.79 |



**Table 3.** Experimental CFD setup.

| Name | Ascending speed (m s$^{-1}$) | Horizontal wind speed (m s$^{-1}$) | Pressure (hPa) | Time integration (s) |
|---|---|---|---|---|
| exp-6m | 6 | 0 | 850 | 0.117 |
| exp-5m | 5 | 0 | 850 | 0.140 |
| exp-4m | 4 | 0 | 850 | 0.175 |
| exp-1m | 1 | 0 | 850 | 0.698 |
| exp-50hPa | 5 | 0 | 50 | 0.144 |
| exp-100hPa | 5 | 0 | 100 | 0.142 |
| exp-300hPa | 5 | 0 | 300 | 0.141 |
| exp-500hPa | 5 | 0 | 500 | 0.140 |
| exp-700hPa | 5 | 0 | 700 | 0.140 |
| exp-925hPa | 5 | 0 | 925 | 0.140 |
| exp-h2.5m | 5 | 2.5 | 850 | 0.122 |
| exp-h5m | 5 | 5 | 850 | 0.092 |