# Peer review of "Application of cloud particle sensor sondes to estimating the number concentration of cloud water droplets and liquid water content: Case studies in the Arctic region"

_Atmospheric Measurement Techniques, 2020_

## Author Comment (AC1)

Reply to Reviewer #1

We would like to thank you for providing highly technical comments regarding our manuscript. We substantially modified the content in the revised form and believe the modifications will be satisfactory and suitable for publication. The point-by-point responses (in blue) are made below.

*The CPS is a sensor that is clearly needed for makinng profiles in clouds with upsonds, dropsonds or even teathered balloons. The original description by Fujiwara et al. was a good introduction but left out four very important details that also limit the usefullness of the present paper. 1) What is the intensity profile of the lasr beam across the sample areas of the two detection systems, 2) What is the shape of the elaser beam, 3) What are the actual dimensions of the two parallelagrams and 4) For the collection geometry what do the Mie scattering cross sections look like for water droplets.*

*Lacking this information in the present paper, all of the corrections that are related to the pulse width and the sizing are irrelevant since they all assume a beam shape and laser intensity that is uniform, assumptions that are likely not the case. I am puzzled that readily available software like Zmax was not employed to model the actual optical system. The figures in 3a and b are speculative, in the words of the author. Speculation has no place in a technical article.*

The reviewer may consider that we are the original members who developed the CPS sondes (this is probably because the title of our original manuscript might provide misleading information); however, this is not the case. Note that we have not been involved in the development of the CPS sondes. We contacted the manufacturer (i.e., Meisei Electric, Co. Ltd.) to obtain the technical information.They explained to us that the shape of the laser beam is not uniform as it is but is adjusted to be uniform with Biconvex lends (the reference is not available); the actual dimensions of the two parallelograms are not evaluated by a optical software.

At this moment, it is difficult for us to answer the four questions raised by reviewer #1 in a satisfactory manner, because we have used CPS sondes as an end user and this paper wanted to propose a practical method based on our field experiments. Nevertheless, we substantially revised the paper, including its title, from the viewpoint of a CPS sonde user to clarify our approach and deliver the message to the manufacturer for further development and experiments (e.g., mapping of the sampling area as suggested by the reviewer). Based on CPS sonde measurements of Arctic low-level clouds, we demonstrated that the original approach published in AMT (Fujiwara et al., 2016) should be improved. For this reason, we still believe that AMT is the best platform for reporting our experiences of CPS sonde measurements in the Arctic and then discussing the need for a new correction method.

The structure of this paper has been simplified as (1) Introduction, (2) Experimental designs, (3) Data processing, (4) Comparison with other data sources, (5) Discussion, and (6) Conclusions. The previous Figs. 3 and 7 and their explanations related to the optical

design and sampling area mapping have been deleted because we would like to highlight the observed data, particularly the smaller particle signal widths (PWDs) than those expected by Fujiwara et al. (2016). We also removed some simulated results (previous Figs. 9 and 13) to keep the manuscript simple. Since we realized that additional laboratory experiments would be necessary in future work, their needs have been discussed in the new section by citing literature. The correction factor was newly estimated in section 3.4 by the idea of collection efficiency based on Noll and Pilat (1970), improving the theoretical robustness of estimation of the factor, although the main result was not changed.

*The size calibration is based on water equivalent sizes of crown glass and PSL particles, but these water equivalent sizes have to come from theoretical considerations. Nowhere is this described. The cloud physics community that uses optical spectrometers are now using precise droplet generators to map the sample areas of spectrometers similar to the CPS. This needs to be done for the CPS if this technology is to be accepted and inversions have to be applied as there is no qualifier in the system to constrain the particles through the sample area. I recommend that the authors read the papers related to the IAGOS Backscatter Cloud Probe.*

As suggested by the reviewer, additional laboratory experiments using optical spectrometers and precise droplet generators would provide more realistic data but we consider that it is beyond the scope of this research because the aim of this study is to provide a practical method to correct the CPS data under the existing system from a user perspective. However, we also agree the need for some laboratory experiments as suggested by the reviewer, we cited papers Lance et al. (2010) and Beswick et al. (2014) to introduce how the state-of-art has been applied to calibrating cloud microphysics probes, which must stimulate the manufacturer (i.e., Meisei Electric, Co. Ltd.) for further development of CPS sonde. This issue is discussed in the first paragraph of the new section 5.3. The information on the shape of the laser beam is also mentioned based on personal communication with the manufacturer.

*Finally, trying to model the flow through the CPS with no measurement validation is unconvincing. The much siimpler and more convincing approach is to do the measurements in a low speed wind tunnel that are employed around the world to calibrate anemometers.*

In this paper, we used the data based on the field experiments. We do not conduct laboratory experiments to validate the simulated flow field. Instead, we compared the flow speed data at the bottom of the CPS inlet observed by anemometers in Fujiwara et a. (2016). The flow speed was 15% smaller than the ascending speed. Our simulations also have the same tendency even if the ascending speed is changed from 4 m/s to 6 m/s, suggesting that our simulations are valid for further investigating flow characteristics around the CPS housing. This content has been included in the new section 3.4. This issue is discussed in the second paragraph of the new section 5.3.

*From a presentation perspective of the material, once the study is repeated more vigorously, this is an AMT paper so most of the introduction is irrelevant except for the last paragraph that*

*describes the objectives. The title is misleading and needs to be more explicit and the photos, while pretty, are also irrelevant to the topic.*

We have conducted many CPS sonde measurements in the Arctic region based on the original protocol. As a result, we noticed that the original approach published in AMT (Fujiwara et al., 2016) should be improved. To avoid misleading, we changed the title to the case study in the Arctic. On the other hand, we believe that the scientific background of the role of clouds in global and polar regions is still needed in the introduction.

*Finally, comparisons in the field are irrelevant until the corrections are justified properly and in additions the OPCs against which they are compared have their own uncertainties that have to be explain to put the comparisons into context.*

In the revised form, the correction factor was introduced by the idea of collection efficiency in the new section 3.4 based on Noll and Pilat (1970). The value has been changed from 5.8 to 7.5; however, the main conclusion did not change. We also mentioned the coincidence loss (10%) of the OPC as the product specifications.

References newly included

1. Baumgardner et al. (2017), Cloud ice properties: In situ measurement challenges, Meteorological Monographs, 58, 9.1–9.23.

2. Beswick et al. (2014), The backscatter cloud probe – a compact low-profile autonomous optical spectrometer, Atmospheric Measurement Techniques, 7, 1443–1457.

3. Craig et al. (2013), Design and sampling characteristics of a new airborne aerosol inlet for aerosol measurements in clouds, Journal of Atmospheric and Oceanic Technology, 30, 1123–1135.

4. Lance et al. (2010), Water droplet calibration of the Cloud Droplet Probe (CDP) and inflight performance in liquid, ice and mixed-phase clouds during ARCPAC, Atmospheric Measurement Techniques, 3, 1683–1706.

5. Murakami and Matsuo (1990), Development of the hydrometeor videosonde, Journal of Atmospheric and Oceanic Technology, 7, 613–620.

6. Noll and Pilat (1970), Inertial impaction of particles upon rectangular bodies, Journal of Colloid and Interface Science, 33, 197–207.

---

## Author Comment (AC2)

Reply to Reviewer #2

We would like to thank you for providing highly technical comments regarding our manuscript. We substantially modified the content in the revised form and believe the modifications will be satisfactory and suitable for publication. The point-by-point responses (in blue) are made below.

Overview of the revision

The reviewer may consider that we are the original members who developed the CPS sondes (this is probably because the title of our original manuscript might provide misleading information); however, this is not the case. Note that we have not been involved in the development of the CPS sondes. We contacted the manufacturer (i.e., Meisei Electric, Co. Ltd.) to obtain the technical information.They explained to us that the shape of the laser beam is not uniform as it is but is adjusted to be uniform with Biconvex lends (the reference is not available); the actual dimensions of the two parallelograms are not evaluated by a optical software.

We substantially revised the paper, including its title, from the viewpoint of a CPS sonde user to clarify our approach and deliver the message to the manufacturer for further development and experiments (e.g., mapping of the sampling area as suggested by the reviewer). Based on CPS sonde measurements of Arctic low-level clouds, we demonstrated that the original approach published in AMT (Fujiwara et al., 2016) should be improved. For this reason, we still believe that AMT is the best platform for reporting our experiences of CPS sonde measurements in the Arctic and then discussing the need for a new correction method.

The structure of this paper has been simplified as (1) Introduction, (2) Experimental designs, (3) Data processing, (4) Comparison with other data sources, (5) Discussion, and (6) Conclusions. The previous Figs. 3 and 7 and their explanations related to the optical design and sampling area mapping have been deleted because we would like to highlight the observed data, particularly the smaller particle signal widths (PWDs) than those expected by Fujiwara et al. (2016). We also removed some simulated results (previous Figs. 9 and 13) to keep the manuscript simple. Since we realized that additional laboratory experiments would be necessary in future work, their needs have been discussed in the new section by citing literature. The correction factor was newly estimated in section 3.4 by the idea of collection efficiency based on Noll and Pilat (1970), improving the theoretical robustness of estimation of the factor, although the main result was not changed.

*Major comments*

1. *In order to derive particle concentrations, the information of the sensitive area of the instrument is crucial. The sensitive area is defined as the overlap between the laser beam profile and the detector filed-of-views (FOV). However, no information is given for their geometry. Instead, the authors claim that the "detection domain" is 1x1x0.5 cm based on the dimensions of the slits and in the derivation of total concentration the counts are divided by the cross-sectional area of the CPS inlet (1 cm2), which is not the same as the sensitive area.*

   By the definitions by Fujiwara et al. (2016), the volume of the detection area is 0.5 $cm^3$, while the cross section of the detection area is 1 $cm^2$. We followed the value of the cross section of the detection area to calculate the number concentration. Therefore, "cross-sectional area" has been reworded as "the cross section of the detection area". The results were not changed.

2. *Since the sensitive area is not known, estimation of total concentration is not possible.*

   Because the detailed assessment of the sensitive area is beyond the scope of this study, which the manufacturer would hopefully do, we assessed how to treat and interpret the output of total count by the CPS sonde from an end user perspective. Even if the cross section of the detection area is unknown for an end user, we think that the efforts to validate the total concentration are still meaningful.

3. *The analysis is based on the assumption that the particles have a constant PSW. In order this to be true, the laser profile should be uniform in the detection area and the flow of the particles should be constant. Why no technical efforts were made to fulfil these conditions (e.g. beam shaping, focusing air flow, etc.)?*

   The analysis by Fujiwara et al. (2016) was based on the assumption that the particles have a constant PSW. However, we were not based on the assumption because our field measurements indicated much lower PSW values. This was a central motivation of our study; however, the optical assessment and consideration were removed (previous Figs. 3 and 7) because we focused on the observational results from the standpoint of a CPS sonde user. In addition to this, we communicated with the manufacturer to obtain information on the shape of the laser beam. The issue has been included in the new section 5.3.

4. *The size calibration is based on standard particles. Although the differences in refractive index is taken into account, the authors could have repeat the calibration using water droplets distributed with piezo-injector, which is the common practise for cloud instruments. Same method could have been used to map the sensing area.*

   The additional laboratory experiments using optical spectrometers and precise droplet generators would provide more realistic data but are beyond the scope of this research because the aim of this study is to provide the practical method to correct

the CPS data under the existing system from a user perspective. Instead of this, we cited papers Lance et al. (2010) and Beswick et al. (2014) to introduce how the state-of-art has been applied to calibrating cloud microphysics probes, which must stimulate the manufacturer (i.e., Meisei Electric, Co. Ltd.) for further development of CPS sonde. This issue is discussed in the first paragraph of the new section 5.3.

5. *According to flow calculations, the flow speed at the inlet is reduced by 17.2% due to the instrument housing. At the same time air pressure increases. After reaching the instrument, the flow speed accelerates to a value close to the flow speed before the instrument. The authors interpret these calculations so that "the chance that the air mass can enter the CPS inlet in a unit of time is reduced to 17.2%" and calculate a correction factor for the total counts to be 5.8 (= 1/0.172). However, I consider this reasoning to be incorrect.*

   In this paper, we used the data based on the field experiments. We do not conduct laboratory experiments to validate the simulated flow field. Instead, we compared the flow speed data at the bottom of the CPS inlet observed by anemometers in Fujiwara et a. (2016). The flow speed was 15% smaller than the ascending speed. Our simulations also have the same tendency even if the ascending speed is changed from 4 m/s to 6 m/s, suggesting that our simulations are valid for further investigating flow characteristics around the CPS housing. This content has been included in the new section 3.4. Then, the correction factor was introduced by the idea of collection efficiency in the new section 3.4 based on Noll and Pilat (1970), improving the theoretical robustness of estimation of the factor, although the main result was not changed.

*Minor comments*

1. *p.2, lines 49-52: Cloud phase can be determined in-situ using number of different methods but the authors only mention Cloud Particle Imager. I would suggest referring to paper by Baumgardner et al., 2017.*

   Thank you for introducing the paper. We added Baumgardner et al. (2017), Lance et al. (2010) and Beswick et al. (2014) in the introduction.

2. p.4, line 93: The terms "particle signal width" (PSW) and particle transit time are used. Why not to use the term particle time-of-flight (TOF) that is frequently used in the community?

   We'd like to just follow this term (PSW) from Fujiwara et al. (2016, AMT) for the consistency and readability in AMT papers.

3. p.5, Section 2.5: No explanation is given for the chosen DOP threshold. Additionally, the separation is based on particle sphericity rather than actual ice/water phase. This should be mentioned.

The threshold of 0.3 was originally proposed by Fujiwara et al. (2016) based on laboratory experiments using standard particles; however, they also showed that the DOPs for liquid clouds were usually higher than 0.5 in actual observations (Figs. 4a, 7a, 10a in Fujiwara et al. (2016)). Because the mixed-phase clouds are typical form in the Arctic, the value of 0.5 would be more suitable than 0.3 to reduce the chance of counting ice particles as liquid particles. This explanation has been included in the section 2.4.

4. Fig. 6: I don't see that accumulated relative PSW frequency is a good way to illustrate the PSW distribution. Why not to show normalised PSW frequency? Why is the data limited to water cases (DOP>0.5)?

   The accumulative relative frequency of PSW is a suitable indicator to demonstrate how PSW takes a smaller values than 1.0 ms.

5. Fig. 10: What is the upper detection limit of the OPC?

   Unfortunately, the detection limit is unknown by the instrument specification.

6. Fig. 10b: Why is the OPC counting particles with concentration >100 L-1 below the cloud base?

   These are not cloud particles but aerosol particles.

References newly included

1. Baumgardner et al. (2017), Cloud ice properties: In situ measurement challenges, Meteorological Monographs, 58, 9.1–9.23.

2. Beswick et al. (2014), The backscatter cloud probe – a compact low-profile autonomous optical spectrometer, Atmospheric Measurement Techniques, 7, 1443–1457.

3. Craig et al. (2013), Design and sampling characteristics of a new airborne aerosol inlet for aerosol measurements in clouds, Journal of Atmospheric and Oceanic Technology, 30, 1123–1135.

4. Lance et al. (2010), Water droplet calibration of the Cloud Droplet Probe (CDP) and inflight performance in liquid, ice and mixed-phase clouds during ARCPAC, Atmospheric Measurement Techniques, 3, 1683–1706.

5. Murakami and Matsuo (1990), Development of the hydrometeor videosonde, Journal of Atmospheric and Oceanic Technology, 7, 613–620.

6. Noll and Pilat (1970), Inertial impaction of particles upon rectangular bodies, Journal of Colloid and Interface Science, 33, 197–207.

---

## Author Comment (AC3)

Reply to Reviewer #3

We would like to thank you for providing highly technical comments regarding our manuscript. We substantially modified the content in the revised form and believe the modifications will be satisfactory and suitable for publication. The point-by-point responses (in blue) are made below.

Overview of the revision

The reviewer may consider that we are the original members who developed the CPS sondes (this is probably because the title of our original manuscript might provide misleading information); however, this is not the case. Note that we have not been involved in the development of the CPS sondes. We contacted the manufacturer (i.e., Meisei Electric, Co. Ltd.) to obtain the technical information.They explained to us that the shape of the laser beam is not uniform as it is but is adjusted to be uniform with Biconvex lends (the reference is not available); the actual dimensions of the two parallelograms are not evaluated by a optical software.

We substantially revised the paper, including its title, from the viewpoint of a CPS sonde user to clarify our approach and deliver the message to the manufacturer for further development and experiments (e.g., mapping of the sampling area as suggested by the reviewer). Based on CPS sonde measurements of Arctic low-level clouds, we demonstrated that the original approach published in AMT (Fujiwara et al., 2016) should be improved. For this reason, we still believe that AMT is the best platform for reporting our experiences of CPS sonde measurements in the Arctic and then discussing the need for a new correction method.

The structure of this paper has been simplified as (1) Introduction, (2) Experimental designs, (3) Data processing, (4) Comparison with other data sources, (5) Discussion, and (6) Conclusions. The previous Figs. 3 and 7 and their explanations related to the optical design and sampling area mapping have been deleted because we would like to highlight the observed data, particularly the smaller particle signal widths (PWDs) than those expected by Fujiwara et al. (2016). We also removed some simulated results (previous Figs. 9 and 13) to keep the manuscript simple. Since we realized that additional laboratory experiments would be necessary in future work, their needs have been discussed in the new section by citing literature. The correction factor was newly estimated in section 3.4 by the idea of collection efficiency based on Noll and Pilat (1970), improving the theoretical robustness of estimation of the factor, although the main result was not changed.

*How does the laser beam profile look like? In optical particle detection, the laser beam profile in the sample volume plays an important role. This is a major point that needs to be added to this manuscript. Best practice is to use optical modeling software and also measure the intensity profile at different places within the sample volume with a photodiode on a translation stage or a*

*camera.*

We communicated with the manufacturer to obtain the information on the shape of the laser beam (the reference is not available). The manufacturer did not conduct the optical modeling, but the shape of the laser beam is adjusted to be uniform by using the Biconvex lens. The issue has been included in the new section 5.3.

*Another point that deserves more investigation is the correction factor for the total particle counts. The authors already mentioned inhomogeneities in the sample volume, so one of the basic assumptions is actually violated. In my opinion, this issue can only be resolved by intercomparison with state-of-the-art optical cloud particle spectrometers in a laboratory setup.*

As suggested by the reviewer, additional laboratory experiments using optical spectrometers and precise droplet generators would provide more realistic data but we consider that it is beyond the scope of this research because the aim of this study is to provide a practical method to correct the CPS data under the existing system from a user perspective. However, we also agree the need for some laboratory experiments as suggested by the reviewer, we cited papers Lance et al. (2010) and Beswick et al. (2014) to introduce how the state-of-art has been applied to calibrating cloud microphysics probes, which must stimulate the manufacturer (i.e., Meisei Electric, Co. Ltd.) for further development of CPS sonde. This issue is discussed in the first paragraph of the new section 5.3. The correction factor was introduced by the idea of collection efficiency in the new section 3.4 based on Noll and Pilat (1970). The value has been changed from 5.8 to 7.5; however, the conclusion did not change.

*Although the flow conditions have been investigated via CFD modeling, there are no results from real measurements shown. In particular, boundary layer effects and "slow flow zones" are much clearer to see in an experimental flow characterization in a wind tunnel. Ideally, a particle generator is part of the laboratory setup to also investigate how the flow conditions influence detectability of cloud hydrometeors. In addition, a particle generator producing water droplets should be used to calibrate the CPS sondes. Further experiments in an icing wind tunnel would be helpful to investigate the ability of the sensor to distinguish ice from supercooled liquid water under realistic conditions.*

In the revised form, we compared the simulated flow speed with the flow speed data at the bottom of the CPS inlet observed by anemometers by Fujiwara et al. (2016). The flow speed was 15% smaller than the ascending speed. Our simulations also have the same rate even if the ascending speed is changed from 4 m/s to 6m/s, suggesting that our simulations are valid for further investigating flow characteristics around the CPS housing. This content has been included in the new section 3.4. Although we do not have the observed data of the "slow flow zones" by using a particle generator, the hypabolic flows would be common feature at the plane surface. Further experiments for distinguishing ice from supercooled liquid water would be desired; however, we think it is beyond the scope of this study.

References newly included

1. Baumgardner et al. (2017), Cloud ice properties: In situ measurement challenges, Meteorological Monographs, 58, 9.1–9.23.

2. Beswick et al. (2014), The backscatter cloud probe – a compact low-profile autonomous optical spectrometer, Atmospheric Measurement Techniques, 7, 1443–1457.

3. Craig et al. (2013), Design and sampling characteristics of a new airborne aerosol inlet for aerosol measurements in clouds, Journal of Atmospheric and Oceanic Technology, 30, 1123–1135.

4. Lance et al. (2010), Water droplet calibration of the Cloud Droplet Probe (CDP) and inflight performance in liquid, ice and mixed-phase clouds during ARCPAC, Atmospheric Measurement Techniques, 3, 1683–1706.

5. Murakami and Matsuo (1990), Development of the hydrometeor videosonde, Journal of Atmospheric and Oceanic Technology, 7, 613–620.

6. Noll and Pilat (1970), Inertial impaction of particles upon rectangular bodies, Journal of Colloid and Interface Science, 33, 197–207.

---

## Author Response (AR2)

Reply to Reviewer #1

We would like to thank you again for providing helpful comments regarding our manuscript. We communicated with the manufacturer regarding the shape of the pulse and added one discussion paragraph to reply to the referee's comments. We hope the content in the second revision will be satisfactory and suitable for publication. The point-by-point responses are made below.

1. *Corrections to the number concentration using pulse width analysis.*

   The first issue lies in the assumptions that go into doing a coincidence correction based on the transit time. The primary assumption assumes that differences in the actual pulse width (PW), compared to the expected, are mainly a result of flow velocity variations or multiple particles in the beam. Hence, the authors go to a lot of effort to model the flow around the sonde and in the particle deliver system but never do they acknowledge that the large variations in PW are like a result of the non-uniform beam intensity and non-uniform beam geometry. Without taking these factors into account, something requires inforrmation and cooperation from the manufacturer, the correction factors that are derived are meaningless. The probability of coincident can be calculated quite directly since the sample volume of the sonde seems to be known. Why isn't this done?

   Regarding the beam characteristics, the manufactures disclosed to us an example of the result of a beam profiler (a captured image: Fig. 1). As we can see, the beam intensity is not necessarily uniform and is weakened outside. The heterogeneity of the laser beam would cause the change of pulse shape. Examples of the pulse shape are shown in Fig. 2. The pulse shape is not a rectangle but a smoothed shape, making a shorter pulse length in smaller voltage (i.e., smaller particle).

   Although we can not manage this issue technically, we can estimate the expected relationship between pulse intensity ($I_{55}$) and pulse length (PSW) and compare the observed results. Now, we assume the pulse shape is not a rectangle but a sine curve due to the heterogeneity of the laser beam. Considering the $I_{55}$ is recorded under condition $> 0.3V$, the following condition would hold:

   $$\frac{I_{55}}{2} \times sin(2\pi(\frac{PSW_o}{PSW_e} + 0.25) + 1) = 0.3 \tag{1}$$

   Here, $PSW_o$ and $PSW_e$ are observed and expected PSW, respectively. As an ideal case, the condition of $PSW_e$=1.0 is considered here. The examples of the pulse shape are shown in Fig. 3. The smaller pulse intensity provides a short pulse length. The relationship between $I_{55}$ and $PSW_o$ derived using Eq. (1) is shown in Fig. 4a (a black line). The data observed in the Arctic regions are mostly on the black line, suggesting that the CPS counted the particles as a single particle in case of the smaller

[Figure]

Figure 1: A capture image of beam intensity in the CPS inlet across the air stream using a beam profiler. (provided by SHINYEI Technology, Co., Ltd.)

[Figure]

Figure 2: Pulse shapes observed by an oscilloscope in a laboratory. (provided by MEISEI Electric, Co. Ltd.)

pulse intensity ($I_{55} < 3$ V). For the larger pulse intensity ($I_{55} > 3$ V), the overlapping would occur but was not significant for our data (Arctic regions).

Because the shorter $PSW_o$ for smaller particles allows counting the more particles in a unit time, we can also estimate the countable particle number in a unit time (e.g., 1 sec). Fig. 4b shows the upper limit for the countable particle numbers as a function of $I_{55}$. If the background number concentration is very low (e.g., 1000 particles s$^{-1}$, a black line in Fig. 4b), then every size can be detected as a single particle. In the case that the concentration is relatively high (e.g., $> 2000$ s$^{-1}$), however, particle overlap is potentially expected for the larger particles. In such a situation, the value of $PSW_o$ can be considered an overlap factor for particle overlap. In our case, the background number concentration is low (typically $<1500$ particles$^{-1}$) and the majority of the observed particles have relatively small sizes; thus, the effect of the overlap factor (around 1.5 or less) on estimating the total count is relatively small compared with

[Figure]

Figure 3: Examples of pulse shapes estimated by Eq. (1).

the effect of collection factor (=7.5).

This discussion has been included in a new section of "5.4 Limitation of CPS sondes."

2. *Poor statistics on particle size and shape.*

The second issues, that of poor statistics on particle size and shape, concerns how effective radius, LWC and shape are derived. The data transmission rate, according to the authors and manufacturer, is 25 bytes/second. The manufacturer has chosen to use this to transmit size and shape information for only the first six detected particles each second. According to the drawings, the sensitive sample area presented to particles in the inlet is 1 cm2. Since the flow velocity is approximate 5 ms-1, this means that the CPD will be detecting approximately 500 cm3 per second. Even if the cloud concentrations are very small, for example 10 cm-3, this will be 5000 particles/second. If the CPD can only transmit size information on the first six out of these 5000 particles, this is only 0.12% of the particles. Statistically speaking, the probability that these 6 particles represent the parent population is less than 1%. This by itself makes the CPD of limited use, but what is even more unfortunate is that the size that is derived from these six particles will be heavily biased toward smaller sizes, give the generally log normal size distribution of droplets in cloud where smaller droplets dominate the number concentration, i.e. the first 6 particles samples will mostly likely be in the smallest droplets. Likewise, in a mixed phase cloud, the liquid phase will predominate so that the polarizatio ratio will be biased

[Figure]

Figure 4: (a) Estimated relationship between PSW and $I_{55}$ (a black line) in case of $PSW_e$ = 1.0. Blue and red lines indicate the overlap factor to correct the count for each situation of the number concentration. Gray dots are the same plot in Fig. 6 of the main text. Green dots indicate the mean state for each second after applying cut-off value of PSW ($PSW_c$). (b) Upper limit of countable particle number as a function of $I_{55}$ in case of $PSW_e$ = 1.0.

toward the droplets rather than ice.

In our observations, the typically observed counts were around 2000 $L^{-1}$, which corresponds to 1000 particles $s^{-1}$. Because the phase-detection depends on the first six particles per second (i.e., 0.6 % of 1000 particles), the representation of size distribution in every second might be insufficient. However, the fact that the corrected number concentration matched with the OPC measurements reveals that the correction method in this study would be applicable for the clouds under relatively low number concentration without particle overlapping. The reason would be related to the collection efficiency of the CPS housing. Considering the collection efficiency of 13.3% derived from section 3.4, the number of expected particles pass across the CPS inlet would be 133 particles $s^{-1}$. Considering that it usually takes tens of seconds to observe a few hundred-meter-thick clouds, it should be noted that each first six particles during the assent are not selectively counted. Assuming the mean state of the clouds in five seconds (i.e., a 25-m thick cloud layer), 30 particles are available for estimating the size and phase of particles (more than 20% of expected particles). This condition represents the total size distribution under a 90% significant level with 10% permissible error. Of course, one should pay attention to the clouds when high

number concentrations and larger particles are expected. In mixed-phase clouds, the liquid phase droplets might predominate due to smaller particles, introducing the biased DOP value toward the droplets rather than ice. Choice of the DOP threshold between ice and liquid is also challenging (in this study, 0.5 was proposed as the DOP threshold). Overall, the instantaneous value obtained by the CPS sonde does not represent the cloud characteristics at the level sufficiently, in particular under relatively higher number concentration with larger droplets; however, the situation under relatively lower number concentration with smaller droplets allows the CPS sondes to measure the mean state of the clouds.

This discussion has been included in a new section of "5.4 Limitation of CPS sondes."

What is puzzling is why the manufacturer chose to waste the limited data transmission by sending individual article information. They could have instead, compiled a size distribution of 5 or 6 channnels, with increasing width. 25 bytes is 200 bits. 12 bits is 4096 counts, so that a size distribution, percentage of ice and some housekeeping could have been encoded much more efficiently thatn the current configuration.

The manufacturer considers updating the system to be able to count more particles efficiently in the near future.

3. Other modifications

Based on the discussion in section 5.4, we modified several sentences in the abstract, conclusion, and Fig. 13.

Reply to Reviewer #3

We would like to thank you for providing helpful comments regarding our manuscript. We substantially modified the content based on Reviewer #1, who asked us the shape of the beam and the resultant effect on the particle counts. We communicated with the manufacturer regarding the shape of the pulse and added one discussion paragraph (section 5.4: Limitation of CPS sondes) to reply to the referee's comments.

Replies to minor comments:

- Subsection 5.1: Were there no satellite overpasses during the field campaign and in particular in spatial and temporal proximity of the balloon launches? Using some satellite remote sensing data to compare with the observations is probably more useful than using a reanalysis from a global circulation model.

  Thank you for the suggestion. Initially, we tried to find opportunities for simultaneous observations with the CloudSat path. However, we were not able to match the satellite path. The validation with other observing systems is, of course, desirable for future campaigns.

- Subsection 5.2: Is there a possibility to put particles of arbitrary size randomly in the CFD modeling data? By putting particles into the flow, it would be possible to elucidate how much the measured data would be affected by sub-isokinetic sampling and if the applied detectability assumptions and the applied correction factor make sense for the actual liquid water path. These numerical experiments are not super complicated and can help a lot to understand how the actual particle size distribution relates to the measured particle size distribution. If the CFD model does not support tracer particles, using the sub-isokinetic flow and its effect on the particles in combination with Monte-Carlo simulations of particles in a volume could be a fallback option to quantify the measurement uncertainties more thoroughly.

  Thank you for providing us an idea to reduce the uncertainties of the collection factor. Unfortunately, our CFD model does not support tracer particles for putting the random size particles. So far, we have concentrated on estimating the collection factor using CFD; however, Reviewer #1 consistently insisted that we confirm the CPS beam geometry and its effect on the particle count. Therefore, an additional discussion paragraph was made in the 2nd revised version. The finding is that there are at least two types of correction factors: the first one is the collection factor related to the CPS housing, the other one is an overlap factor originated from the heterogeneity of the CPS beam. Based on the idealized estimation of pulse intensity ($I_{55}$) and pulse length (PSW), we found that the smaller particles tend to be observed frequently, although the inertial force is smaller than that of larger particles in front of CPS inlet. Therefore, the mixed effect in the observed size distribution makes the further quantification of the collection factor complicated. This would be strong guidance toward the manufacturer for further developing the CPS sondes shortly.

- Other modifications

  Based on the discussion in section 5.4, we modified several sentences in the abstract, conclusion, and Fig. 13.